# Prevalence of stunting and its determinants among children under five in 35 Sub-Saharan African countries (2011–2024): Insights from recent demographic health survey data using a generalized linear mixed-effects model with robust poisson regression

Abdulkerim Hassen Moloro[1]*, Kebede Gemeda Sabo[1], Kusse Urmale Mare[1], Beriso Furo Wengoro[2], Eshetu Elfios Endrias[3], Roda Mehadi Ibrahim[4], Teshager Dubie[5], Oumer Abdulkadir Ebrahim[6], Begetayinoral Kussia Lahole[7]

1 Department of Nursing, College of Medicine and Health Sciences, Samara University, Samara, Ethiopia, 2 Department of Biomedical Sciences, College of Medicine and Health Sciences, Samara University, Samara, Ethiopia, 3 Department of Nursing, School of Nursing, College of Medicine and Health Sciences, Wolaita Sodo University, Wolaita Sodo, Ethiopia, 4 Department of midwifery, College of Medicine and Health Sciences, Jigjiga University, Jigjiga, Ethiopia, 5 Department of Veterinary Medicine, College of Veterinary Medicine and Animal Sciences, Samara University, Samara, Ethiopia, 6 Department of Public Health, College of Medicine and Health Sciences, Samara University, Semera, Ethiopia, 7 Department of Midwifery, College of Medicine and Health Sciences, Arba Minch University, Arba Minch, Ethiopia

* habdulkerim4@gmail.com

## Abstract

### Background

Despite efforts by initiatives like the World Bank's 'All Hands-on Deck' and UNICEF's programs to address stunting through multisectoral approaches, the burden of stunting remains alarmingly high in sub-Saharan Africa. This study utilized recent large-scale survey data from 35 SSA countries (2011–2024) to estimate the pooled prevalence of stunting and its determinants among children under 5 years of age. Key variables such as antenatal care visits, postnatal care, and maternal nutritional indicators, which previous studies did not account for, are incorporated into the analysis.

### Methods

A secondary analysis was conducted using recent demographic and health survey data (2011–2024) from 35 sub-Saharan African (SSA) countries. A total weighted sample of 191,953 children under 5 years of age was included in the analysis. Descriptive and inferential analyses were performed using STATA 17. Forest plots were utilized to illustrate both pooled and country-specific stunting rates. Determinants of stunting were identified through a multilevel mixed-effects Poisson

**Data availability statement:** The datasets analyzed during current study are publicly available from the Demographic and Health Survey (DHS) program, managed by ICF international. The data can be accessed upon reasonable request through the DHS program website at http://www.dhsprogram.com. Researcher must register and submit a data access request through the site. For inquires regarding data access, please contact the DHS Data Archivist at archive@dhsprogram.com. The data used in this study are cited as: ICF. Demographic and Health Surveys (various years). Funded by USAID. Rockville, Maryland. Available at: http://dhsprogram.com.

**Funding:** The author(s) received no specific funding for this work.

**Competing interests:** The authors have declared that no competing interests exist.

**Abbreviations:** AIC, Akaike's Information Criteria; APR, Adjusted Prevalence Ratio; BIC, Bayesian Information Criteria; CI, Confidence Interval; CPR, Crude Prevalence Ratio; DHS, Demographic and Health Survey; ICC, Intra Class Correlation Coefficient; IRB, Institutional Review Board; LL, Log-Likelihood; MOR, Median Odds Ratio; PCV, Proportional Change in Variance; SSA, Sub-Saharan Africa.

regression model with robust variance. The adjusted prevalence ratios and their 95% confidence intervals were used to assess the strength and statistical significance of associations.

## Result

The pooled prevalence of stunting among children under 5 years of age in 35 sub-Saharan African countries was 29.89% (95% CI: 26.63, 33.14%), with the lowest level in Gabon (13.91%) and the highest in Burundi (55.80%). Being male children (aPR = 1.24, 95% CI: 1.21–1.26), being aged 12 months or older (aPR: ≥ 1.81, p < 0.01), insufficient antenatal care (ANC) visits (aPR: ≥ 1.17, p < 0.01), lack of post-natal visits(aPR = 1.03, 95% CI: 1.07, 1.05), children perceived as small or average at birth (aPR: ≥ 1.16, p < 0.01), mother without a higher education (aPR: ≥ 1.94, p < 0.01), living in a poor or average wealth household (aPR: ≥ 1.23, p < 0.01) were significant predictors of stunting. Conversely, maternal overweight (aPR = 0.81, 95% CI: 0.77–0.84) and obese mothers (aPR = 0.88, 95% CI: 0.85–0.90) were associated lower prevalence of stunting.

## Conclusion

Study revealed significant country-level variations and rates exceeding 30% in 15 countries, signaling a major public health concern. The key individual, household and contextual factors associated with stunting in this study suggest the need for immediate actions expanding antenatal and postnatal care, promoting facility-based deliveries, enhancing maternal education, and media outreach. Long term strategies must tackle poverty, food systems, and equitable nutrition access, supported by governance and stability. A multisectoral approach integrating health, education, agriculture, WASH, and social protection ensures substantiable child growth, complemented by longitudinal research for policy coherence.

## Introduction

Stunting defined as a length/height-for-age measurement below −2 Standard Deviations from the median [1], remains a critical and persistent public health crisis in sub-Saharan Africa (SSA). It is the result of chronic malnutrition, a condition that compromises physical function and the maintenance of key bodily processes [2], and it leads to significant impairments in physical, psychological, and cognitive growth [3].

Sub-Saharan Africa carries a disproportionately heavy burden of childhood stunting, with pooled data from 35 countries indicating an pooled prevalence of 35% [4]. National disparities are remarkable, ranging from as low as 17% in Gabon to as high as 59% in Burundi [5], while Nigeria reports a prevalence of 36.5% among children under five [6]. This situation persists despite a global decline in stunting rates, as the absolute number of affected children in Africa has risen [7], underscoring a regional emergency within the worldwide context of 161 million stunted children [8].

The consequences in SSA are severe and lifelong, including increased child morbidity and mortality, impaired development, and higher disease risk [9,10]. Research has identified that stunting arises from a wide range of interrelated causes, which can be categorized into individual-level, household, and contextual factors. At the individual level, determinants include the child's sex and age, maternal age and educational attainment, and birth interval [11,12]. Additional contributors such as birth order [13], antenatal care utilization [14], and duration of exclusive breastfeeding [13,15–19] have also been identified. Furthermore, studies report that inadequate food intake, low perceived birth weight, maternal body mass index (BMI), and poor sanitation significantly increase the risk of childhood stunting [20–22].

Household-level determinants encompass household size, particularly households with multiple under-five children [14,16,23], access to improved drinking water sources [15], and availability of toilet facilities [24] and household wealth index [23,25]. Contextual factors including place of residence [14] and geographical sub-region [11,12] further affects child nutrition outcomes, underscoring the multifaceted nature of stunting and its dependence on both biological and socio-environmental conditions(Fig 1).

Stunting remains a major child health challenge in sub-Saharan Africa (SSA), despite decades of global and regional initiatives aimed at reducing malnutrition. Programs such as the World Bank's "All Hands-on Deck" initiative, which promotes multisectoral strategies across agriculture, education, social protection, and WASH sectors, and UNICEF's maternal and child nutrition programs have sought to accelerate progress [26,27]. Nevertheless, the burden of the stunting in SSA continues to be alarming high, and many countries are unlikely to achieve the Sustainable Developmental Goal (SSD) 2.2 target of reducing stunting by 40% by 2025 and eliminating all forms of malnutrition by 2030 [28,29].

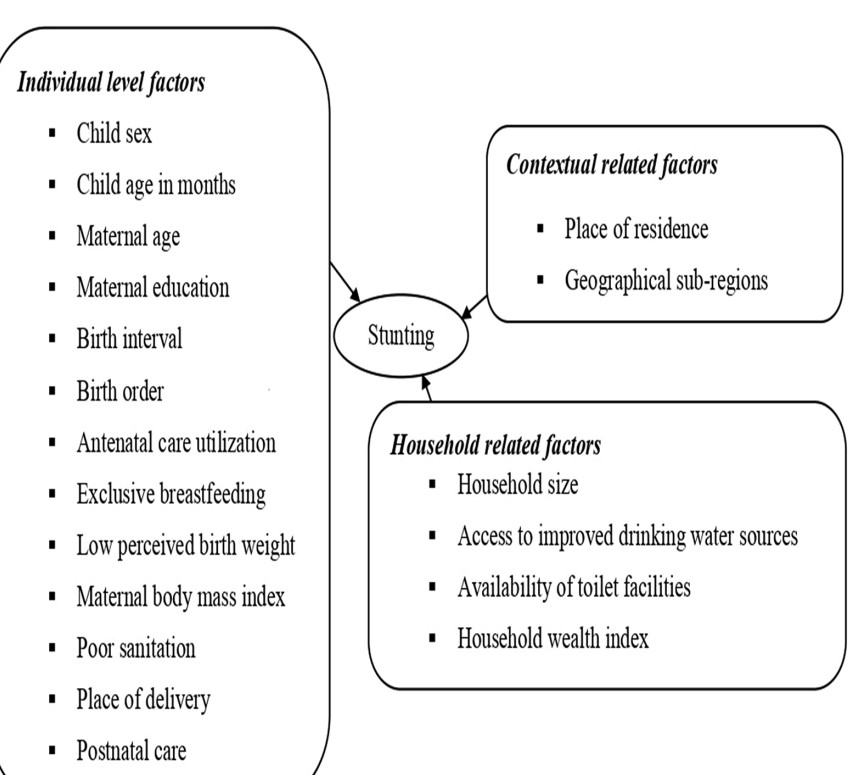

**Fig 1. A conceptual framework showing link between stunting and its determinants among children under 5 years of age, 2011-2024.**

While single country and regional studies have provided valuable insights, their scope is often limited, making it difficult to capture the broader patterns and disparities across the content. This study addresses that gap pooling nationally representative data from 35 SSA countries collected between 2011 and 2024. By combining data at this scale, the analysis enhances statistical power using Poisson regression with robust error variance [30,31] to reduce potential overestimation of odds ratios for common binary outcomes, improves generalizability, and allows for the identifications of common risk factors and cross-country variations that may not be visible in isolated studies.

Additionally, this analysis incorporated key variables such as antenatal care (ANC) visits during pregnancy, postnatal care visits, and maternal nutritional status indicators including BMI and height factors that were not considered in previous studies [4]. By integrating these variables, the pooled evidence is uniquely positioned to inform regional policy, guide resource allocation, and support collaborative interventions across countries. This study therefore provides a more comprehensive and representative picture of stunting in sub-Saharan Africa (SSA), underscoring the urgent need for coordinated, evidence-based strategies to address this persistent public health challenge. Accordingly, the study aimed to determine the pooled prevalence and key determinants of stunting among children under 5 years of age in 35 SSA countries, utilizing generalized linear mixed-effects modeling with robust Poisson regression analysis based on the most recent 2011–2024 Demographic and Health Survey (DHS) data

## Conceptual frame work

## Methods

### Study area, design, and period

This study conducted secondary analysis using data from the Demographic and Health Surveys (DHS) of 35 countries in Sub-Saharan Africa (SSA). The countries involved were Angola, Burkina Faso, Benin, Burundi, DR Congo, Republic of the Congo, Côte d'Ivoire, Cameroon, Ethiopia, Gabon, Ghana, Gambia, Guinea, Kenya, Comoros, Liberia, Lesotho, Madagascar, Mali, Mauritania, Malawi, Mozambique, Nigeria, Niger, Namibia, Rwanda, Sierra Leone, Senegal, Chad, Togo, Tanzania, Uganda, South Africa, Zambia, and Zimbabwe. The selection of country was based on the recent survey year, availability of a standardized and unrestricted dataset, and presence of observations for the outcome variable in the datasets.

The DHS surveys across all countries employed a cross-sectional study design to collect data on basic sociodemographic characteristics and various health indicators, including nutrition, maternal, and child health. For the current analysis, we included the countries that have their recent DHS conducted between 2011 and 2024. Since its inception, DHS has been conducted in over 85 low- and middle-income countries (LMICs) worldwide.

### Population, sampling technique and weight

The source population for this study comprised all children under five years old in 35 SSA countries. The study population included all children under 5 years of age in the survey. Across all countries, the surveys used a multistage stratified cluster sampling technique to select the study participants. In the first stage, each country was divided into clusters, and clusters were randomly selected based on the probability proportional to their contribution to overall country's population. In the second stage, using the housing census as a sampling frame, a representative number of households was selected from each cluster.

To account for the complex survey design, non-response, and to ensure comparability across countries while preventing any single country or survey year from disproportionally influencing the pooled estimates, DHS sampling weights were applied as recommended. The weight variable (v005) was rescaled by dividing by 1,000,000 and incorporated into all analyses using the *svyset* command in Stata, accounting for primary sampling units (v021) and strata (v022).

This produced a weighted analytic sample of **191, 953** children aged 0–59 months with complete data on the variables of interest (Table 1). Additionally, sensitivity analyses were conducted to assess the influence of individual countries by systematically excluding one country at a time and recalculating the pooled prevalence. This approach adjusts for unequal probabilities of selection and non-response within each survey, ensuring that each country's contribution to the pooled prevalence is proportional to its population size.

**Table 1. Survey years and sample sizes of children under 5 years of age from 35 Sub-Saharan African (SSA) countries included in the study, from 2011 to 2024.**

| Country | Survey year | Unweighted sample size | Weighted sample size |
|---|---|---|---|
| Angola | 2023-2024 | 5,039 | 4,800 |
| Burkina Faso | 2021 | 5,731 | 5,696 |
| Benin | 2017-2018 | 11,670 | 11,717 |
| Burundi | 2016-2017 | 6,048 | 6,222 |
| DR Congo | 2023-2024 | 10,745 | 9,734 |
| Congo | 2011-2012 | 4,475 | 3,859 |
| Cote d'Ivoire | 2021 | 4,779 | 4,300 |
| Cameroon | 2018-2019 | 4,497 | 4,707 |
| Ethiopia | 2019 | 5,076 | 4,972 |
| Gabon | 2019-2021 | 5,343 | 5,009 |
| Ghana | 2022-2023 | 4,405 | 4,051 |
| Gambia | 2019-2020 | 3,811 | 3,511 |
| Guinea | 2018 | 3,466 | 3,374 |
| Kenya | 2022 | 17,327 | 15,336 |
| Comoros | 2012 | 2,387 | 2,468 |
| Liberia | 2019-2020 | 2,445 | 2,166 |
| Lesotho | 2023-2024 | 1,093 | 983 |
| Madagascar | 2021 | 5,778 | 5,719 |
| Mali | 2023-2024 | 13,936 | 14,163 |
| Mauritania | 2019-2021 | 9,830 | 9,955 |
| Malawi | 2015-2016 | 5,149 | 5,160 |
| Mozambique | 2022-2023 | 3,733 | 3,858 |
| Nigeria | 2023-2024 | 9,410 | 9,341 |
| Niger | 2012 | 5,576 | 5,112 |
| Namibia | 2013 | 1,558 | 1,438 |
| Rwanda | 2019-2020 | 3,809 | 3,918 |
| Sierra Leone | 2019 | 4,136 | 4,044 |
| Senegal | 2023 | 4,476 | 4,258 |
| Chad | 2014-2015 | 9,826 | 9,805 |
| Togo | 2013-2014 | 3,185 | 3,058 |
| Tanzania | 2022 | 4,807 | 4,866 |
| Uganda | 2016 | 4,423 | 4,350 |
| South Africa | 2016 | 1,113 | 1,080 |
| Zambia | 2024 | 3,810 | 3,710 |
| Zimbabwe | 2015 | 4,957 | 5,212 |
| Total | | **197,044** | **191,953** |

## Data source

The data for this study were obtained from the DHS women's questionnaire, focusing on children under five years of age (under 60 months), and from the Kids Record dataset (KR file) across 35 countries. All datasets were sourced directly from the official Demographic and Health Surveys (DHS) program website (https://dhsprogram.com/).

## Data extraction and management of missing observations

Prior to data extraction, we identified countries with DHS datasets from surveys conducted between 2011 and 2024, as our aim was to include only those with data collected during this timeframe. A standardized data collection tool and face-to-face interview were used to collect the survey's data. To construct a pooled dataset, we first extracted variables relevant to the study from the data of the children under 5 years of age (under 60 months) from the Kids Record dataset (KR file) across 35 countries. The merged datasets from 35 sub-Saharan African countries were then appended by survey year, creating a pooled datasets for regional analysis of child stunting. Country identifiers and survey weights were retained to account for sampling design and cross-country differences. Variables are then recoded and categorized consistently across all 35 DHS surveys using the Guide to DHS statistics to ensure compatibility.

The handling of anthropometric data in this analysis adhered to the standardized protocols outlined in the Guide to DHS Statistics [32]. Children were excluded or dropped from the analyses of all anthropometric indices if they met any of the following criteria: (i) they were not weighed or measured during the survey, (ii) their recorded weight or height measurements were missing, (iii) their month or year of birth was unknown or missing (rendering accurate age calculation impossible for indices requiring it), or (iv) their derived height-for-age z-scores (HAZ) were flagged as biologically implausible (e.g., outside the range of −6 to +6 SD). These exclusions applied uniformly to both the denominator and numerator in prevalence analysis, constituting a complete-case analysis.

For the independent variables, complete case analysis was employed, retaining only observations with non-missing values for all covariates included in the final model. These steps enhance the transparency and validity of our findings in light of potential cross-country variations in data completeness. A detailed explanation of the DHS methodology, guidelines, and procedures for handling missing data is available on the DHS website [33].

## Variables and measurements

**Dependent Variable. Stunting (height-for-age z-scores):** height for age of children dichotomized as normal (not stunted) if height for age ≥ −2 SD and stunted if height of age < −2 SD form WHO child growth reference ((height/age standard deviation (new who)). Weight measurements were obtained using light weight SECA mother-infant scales with a digital screen designed and manufactured under the guidance of UNICEF. Height measurements were obtained using the Shorr measuring board. Children younger than 24-months were measured for their height while lying down, and children older than 24 months were measured while standing.

**Independent Variables:** A total of 23 variables were incorporated into this study, selected through a systematic process that considered both their availability in the Demographic and Health Survey (DHS) and their documented associations with childhood stunting in prior research [34–36]. Each variable was chosen for its relevance to the study objectives and its role as a known or hypothesized determinants of stunting reflecting influences at individual, household, and contextual level. This study also included variables often overlooked in earlier work, such as antenatal care (ANC) visits during pregnancy postnatal care utilizations, and body mass index.

These variables were categorized into individual-level factors, household factors, and contextual factors. The individual-level factors included the sex of the child (male and female), age of the child in month, birth order (1, 2–4, and 5+), breastfeeding (still breastfeeding, ever breastfed and never breastfed) and perceived size at birth (large, average, and small). Other individual-level factors were maternal educational level (no formal education, primary, secondary, and higher), current maternal working status (yes and no), number of antenatal care visits during pregnancy (0, 1–3, and 4 or more)

and postnatal checks within 2 months (yes and no). Variables such as maternal age (15 –24 and 25–49), age at first birth, antenatal visits during pregnancy (0, 1–3, and 4 or more), place of delivery (home, health facility, other), marital status (single and married) and maternal nutritional status were also considered.

Household factors included household size (small, medium, and large), exposure to media (yes and no), and wealth index (poor, middle, rich). Contextual factors encompassed the place of residence (urban and rural) and geographical sub-regions (West, East, Central, and Southern).

Maternal nutritional status, described using Body Mass Index (BMI) according to the WHO adult BMI classification [37]. BMI was calculated by dividing each woman's weight in kilograms by the square of her height in meters (kg/m²). The survey utilized an electronic SECA 874 flat scale for weight measurements, specially designed for mobile use, and a shore measuring board for height. For analysis, women were categorized into four nutritional status groups based on their BMI: underweight (BMI < 18.5 kg/m²), normal weight (BMI 18.5–24.9 kg/m²), overweight (BMI 25.0–29.9 kg/m²), and obesity (BMI ≥ 30.0 kg/m²) [38].

### Data management and statistical analysis

Stata version 17 was utilized for data cleaning and analysis. Prior to the analysis, the presence of the outcome variable in the DHS dataset for each country was confirmed. All the variables considered in the study were reviewed for missing values. Subsequently, the datasets from 35 SSA countries were appended and weighted to maintain sample representativeness and obtain reliable estimates and standard errors. The pooled prevalence of child stunting was calculated using weighted data on the number of affected children with outcome variable and the total number of study participants in each country included in the analysis. The Stata command for meta-analysis "metan" was executed to present the country-specific and pooled estimates with 95% CI in a forest plot. To explore potential sources of heterogeneity, meta-regressions were conducted [39]. Additionally, sensitivity analyses were performed to evaluate the influence of individual country on the overall pooled estimates [39].

A multilevel mixed-effects Poisson regression model with robust error variance was fitted to identify determinants of stunting among children under five in SSA. This model was selected for two primary reasons. First, it directly estimates prevalence ratios, which are more interpretable and avoid the overestimation of association strength common with odds ratios from logistic regression when applied to common binary outcomes in cross-sectional data [30,31]. Second, its multilevel framework accounts for the hierarchical structure of the DHS data, where children are nested within households, and households within clusters, by including random intercepts at the household and cluster levels. Bivariable multilevel robust Poisson regression analysis was done and all variables with a p-value of less than 0.25 in this analysis were considered for multivariable multilevel robust Poisson regression model [30,31].

Prior to model fitting, we assessed the key assumption of the Poisson regression model. A key assumption of this model is that observations are independent and that the mean and variance of the dependent variable are equal. Model adequacy was then confirmed through goodness-of-fit tests: the deviance statistic (151,692.69) and Pearson statistic (83,880.11) both yielded p-values of 1.000, indicating an excellent fit. The data had a mean of 0.306 and a variance of 0.294, producing a mean-to-variance ratio of approximately 1.04. This close alignment supports the equidispersion assumption and validates the use of the Poisson model. Based on these findings, we applied Poisson regression to examine the effects of predictors on stunting, while carefully monitoring for influential observations and residual patterns [40]. Model fit was further assessed using Akaike (AIC) and Bayesian (BIC) Information Criteria, with lower values indicating better fit and parsimony.

In our analysis, five hierarchal models were fitted to select the model that best fits the data: null-model (a model with only outcome variable to assess the random variability in the intercept), model-I (a model with only individual-level explanatory variables), model-II (a model with only household-level explanatory variables), and model-III (a model with contextual-level factors) and model-IV (a model with only potential candidate variables from individual, household, and contextual-level factors). The " meglm" command in Stata was used to fit these models.

Random variability in stunting across clusters was assessed by intra-class correlation coefficient (ICC), explained variance or proportion change in variance (PCV), and median odds ratio (MOR). Akaike's information criteria (AIC), Bayesian information criteria (BIC), Log-likelihood (LL), and deviance (i.e.,-2*LL) values were used for model comparison. The presence of multicollinearity between explanatory variables was checked using variance inflation factor values and the values for the included variables ranged from **1.01 to 1.95,** suggesting that there was no multi-collinearity. Finally, in the multivariable analysis, a p-value less than 0.05 and an adjusted prevalence ratio with the corresponding 95% confidence interval was used to identify the factors associated with stunting among children under 5 years of age in 35 SSA countries.

### Ethical considerations

For this study, we utilized Demographic and Health Survey (DHS) data from 35 sub-Saharan African countries. The DHS survey procedures were approved by the ICF Institutional Review Board (IRB) and the respective host country IRB; therefore, no additional ethical approval was required for this secondary analysis. The dataset accessed contain no identifiable participant information, ensuring confidentiality and privacy. Access to the data was formally authorized by the DHS program, which serve as the institutional custodian of the datasets, through an online request submitted at http://www. dhsprogram.com and supported by authorization letter (AuthLetter_215093) on dated 01/19/2025.

## Results

The study included and analyzed a weighted sample of 191,953 mothers with children aged 0–59 months. Of these children, 50.49% (96,907) were male, and 20.01% (42,185) were under 12 months old. More than half of the children, 55.34% (96,976), had been breastfed at some point but were no longer breastfeeding. Stunting was highly prevalent, affecting 21.98% (42,185) of children under 12 months, compared to older age groups. Among the included children, 65.63% (125,969) lived in rural areas, and 53.29% (25,499) were from households in the poor wealth index category. Additionally, 42.23% (81,068) of the children were from West African countries, and 31.51% (56,237) were delivered at home (Table 2).

Of the 191,953 mothers included in the study, 139,744 (72.80%) were between 25 and 49 years old, and 149,142(77.70%) were currently married. Over one-third of the mothers, 71,108(37.04%), had no formal education, while only 8,824(4.60%) had completed higher education. The majority, 110,643(57.64%), had their first child before the age of twenty, and 126,538 (65.92%) had access to mass media. Regarding nutritional status, 12,591(6.56%) of the mothers were underweight, while 91,612(47.73%) had a body mass index (BMI) within the normal range. Additionally, 26,149 (13.62%) were overweight, and 61,600(32.09%) were classified as obese

### Pooled prevalence of stunting among children under 5 years of age in sub–Saharan African

The pooled prevalence of stunting among children under 5 years of age in 35 sub-Saharan African (SSA) countries was 29.89% (95% CI: 26.63, 33.14%), with significant variation observed across countries ($I^2 = 99.6$, P-value = 0.000) (Fig 2). Gabon had the lowest stunting prevalence at 13.91%, while Burundi reported the highest at 55.80%. Among the 35 countries analyzed, 15 had a stunting prevalence of 30% or higher among children under 5 years of age.

### Handling heterogeneity

The random-effects model revealed considerable heterogeneity. To address this, sensitivity analysis, subgroup analysis, and meta-regression were conducted.

### Sensitivity analysis

Sensitivity analysis was performed to evaluate the effect of individual country on the pooled estimated. When individual country was omitted, the pooled prevalence obtained was within the 95% CI of the overall pooled prevalence. This confirms the absence of single study impact on the overall pooled effect size. Therefore, from the random effects model, there were no country that excessively influence the overall pooled estimate of stunting (Fig 3).

**Table 2. Sociodemographic characteristic of the children under 5 years of age by stunting and their overall background status, in 35 SSA countries, 2011–2024 (n = 191,953).**

| Characteristics | Weighted Frequency (%) | Stunting |
|---|---|---|
| **Sex of the child** | | |
| Male | 96,907(50.49) | 31,604(32.61) |
| Female | 95,045(49.51) | 26,642(28.03) |
| **Age of the child (months)** | | |
| Less than 12 months | 42,185(21.98) | 7,156(16.96) |
| 12–23 months | 39,907(20.79) | 13,492(33.81) |
| 24–35 months | 36,961(19.26) | 14,225(38.49) |
| 36–47 months | 37,708(19.64) | 13,108(34.76) |
| 48–59 months | 35,190(18.33) | 10,264(29.17) |
| **Birth order** | | |
| First | 41,755(21.75) | 11,640(27.88) |
| 2–4 | 94,840(49.41) | 27,615(29.12) |
| 5 or higher | 55,358(28.84) | 18,991(34.31) |
| **Breastfeeding** | | |
| Ever breastfeed | 96,976(55.34) | 34,331(35.40) |
| Never breastfeed | 6,870(3.92) | 2,149(31.28) |
| Still breastfeeding | 71,397(40.74) | 18,261(25.58) |
| **Perceived size at birth** | | |
| Large | 53,478(32.87) | 15,044(28.13) |
| Average | 81,999(50.40) | 25,391(30.97) |
| Small | 27,219(16.73) | 10,537(38.71) |
| **Mothers' educational status** | | |
| No education | 71,108(37.04) | 25,867(36.38) |
| Primary | 61,015(31.79) | 20,175(33.07) |
| Secondary | 51,005(26.57) | 11,385(22.32) |
| Higher | 8,824(4.60) | 818(9.28) |
| **Mother currently working** | | |
| Yes | 109,618(77,227) | 33,980(31.00) |
| No | 77,227(41.33) | 22,379(28.98) |
| **Mother's age (years)** | | |
| 15-24 | 52,209(27.20) | 16,514(31.63) |
| 25-49 | 139,744(72.80) | 41,732(29.86) |
| **Number of ANC visits** | | |
| No visit | 12,189(6.35) | 4,919(40.36) |
| 1 to 3 visits | 39,646(20.65) | 12,762(32.19) |
| 4 and above visit | 140,117(73.00) | 40,565(28.95) |
| **Place of delivery** | | |
| Home | 56,237(31.51) | 16,649(38.46) |
| Health facility | 122,213(69.64) | 38,187(61.54) |
| Postnatal care visits | | |
| Yes | 44,774(34.71) | 11,787(26.33) |
| No | 84,227(65.29) | 25,394(30.15) |
| **Marital status** | | |
| Single | 42,811(22.30) | 12,499(29.20) |
| Married | 149,142(77.70) | 45,747(30.67) |

*(Continued)*

**Table 2.** (Continued)

| Characteristics | Weighted Frequency (%) | Stunting |
|---|---|---|
| **Mothers age at 1st birth** | | |
| Less than 20 | 110,643(57.64) | 36,334(32.84) |
| 20 and above | 81,309(42.36) | 21,912(26.95) |
| **Maternal nutritional status** | | |
| Underweight | 12,591(6.56) | 5,131(40.76) |
| Normal weight | 91,612(47.73) | 31,342(34.21) |
| Overweight | 26,149(13.62) | 6,036(23.08) |
| Obese | 61,600(32.09) | 15,736(25.55) |
| **Household size** | | |
| Small | 16,665(8.68) | 4,882(29.30) |
| Medium | 12,723(6.63) | 3,719(29.23) |
| Large | 162,563(84.69) | 49,644(30.54) |
| **Mass media exposure** | | |
| No | 65,415(34.08) | 25,077(38.34) |
| Yes | 126,538(65.92) | 33,169(26.21) |
| **Wealth index** | | |
| Poor | 84,032(43.78) | 25,499(53.29) |
| Middle | 38,451(20.03) | 11,668(20.53) |
| Rich | 69,471(36.19) | 21,080(28.18) |
| **Place of residence** | | |
| Urban | 65,983(34.37) | 14,229(21.57) |
| Rural | 125,969(65.63) | 44,017(34.94) |
| **Subregion** | | |
| West African Countries | 81,068(42.23) | 22,671(27.97) |
| East African Countries | 47,706(24.85) | 14,833(31.09) |
| Central African Countries | 37,024(19.29) | 12,289(33.19) |
| Southern African Countries | 26,154(13.63) | 8,452(26,154) |

## Sub-group analysis

Subgroup analyses were performed by sub-region, and year of publication.

## Stunting by Sub-Region

Subgroup analysis by sub-region indicated that Est African countries had the highest pooled prevalence of stunting among children under 5 years of age (33.73%, 95% CI: 24.19–43.26), followed by Southern African countries (31.37%, 95% CI: 27.20–35.54), Central African countries (29.98%, 95% CI: 18.24–41.71) and West African countries (26.83%, 95% CI: 22.89–30.77). Substantial heterogeneity was observed in East and Central Africa ($I^2 = 99.8\%$, $P = 0.00$), West Africa ($I^2 = 99.4\%$, $P = 0.00$), and Southern Africa ($I^2 = 98.1\%$, $P = 0.00$) (Fig 4).

## Stunting by Year of Survey

Subgroup analysis by year of survey showed that the pooled prevalence of stunting was highest in 2011–2018 (32.32%, 95% CI: 27.19–37.44) and lowest in 2019–2024 (28.45%, 95% CI: 24.57–32.33). Both periods demonstrated significant heterogeneity (2011–2018: $I^2 = 99.5\%$, $p < 0.00$; 2020–2024: $I^2 = 99.6\%$, $p < 0.00$) (Fig 5).

**Fig 2. Forest plot of the country-level and pooled prevalence of stunting among children under 5 years of age across 35 SSA countries, 2011–2024.**

## Meta-regression

A meta-regression was performed to assess whether the year of survey, sample size, country, or sub-region could explain the heterogeneity in stunting prevalence (Table 3). None of these variables were statistically significant predictors of heterogeneity (all p-values > 0.05). The high residual heterogeneity in the pooled prevalence of stunting suggests that other unmeasured factors are responsible for the variability across studies

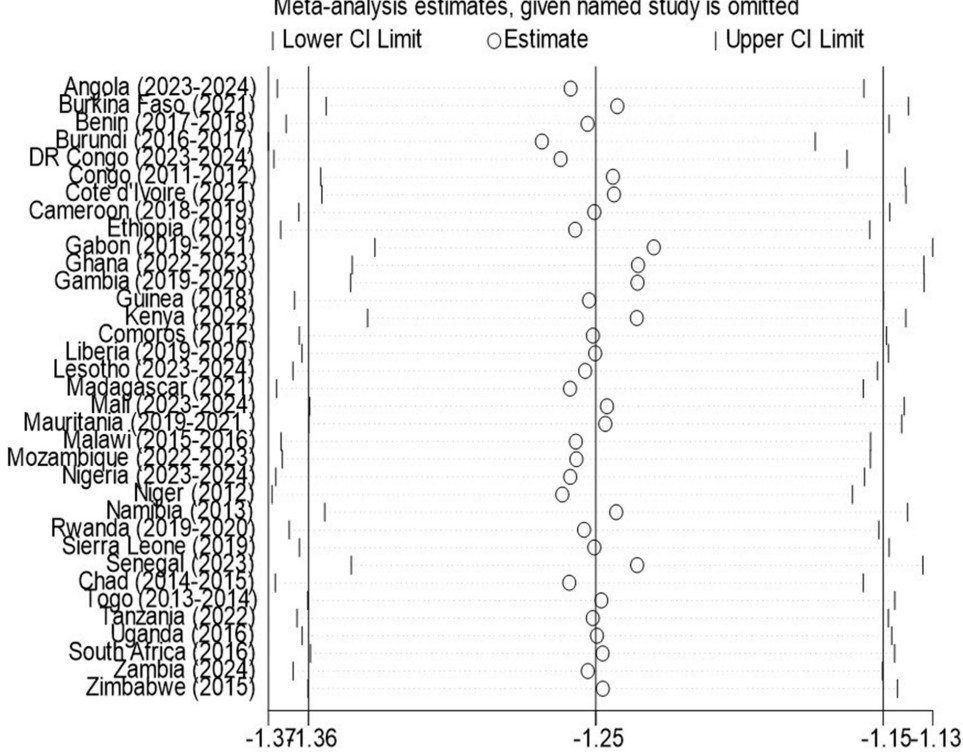

**Fig 3. Sensitivity analysis plot forest plot of the country-level and pooled prevalence of stunting among children under 5 years of age across 35 SSA countries, 2011–2024.**

### Results of random-effects analysis (measures of variation)

The ICC (Intraclass Correlation Coefficient) value from the null model revealed that only about 26.8% of the total variance in stunting among under five children is attributable to differences between clusters. The remaining 73.2% of the variance reflects variation occurring within clusters, which encompasses individual, household, and broader contextual factors. This indicates that while cluster-level differences play a substantial role, the majority of the variability in stunting arises from determinants operating at multiple levels within clusters. In the final model (Model IV), the explained variance values demonstrated that around 47.2% of the total variation in the pooled prevalence of stunting among children under 5 years of age was influenced by the combined effects of individual level factors, household level factors and contextual factors-level factors.

Additionally, the presence of heterogeneity in stunting across clusters was supported by the MOR (Median Odds Ratio) values of 1.96 and 1.46 in the null and full models, respectively. This suggests that children under 5 years of age in clusters with higher pooled prevalence of stunting had approximately 1.96 times greater odds of stunting compared to those in clusters with lower levels of pooled prevalence of stunting. Model 4 was identified as the best-fitted model, as it exhibited the lowest AIC (Akaike Information Criterion), BIC (Bayesian Information Criterion), and deviance values (Table 4).

### Determinants of stunting among children under 5 years of age in 35 sub–Saharan African Countries

Our analysis showed that the prevalence of stunting was 1.24 times higher among male children compared to female children (aPR = 1.24, 95% CI: 1.21–1.26). Using children under 12 months of age as the reference category, the prevalence

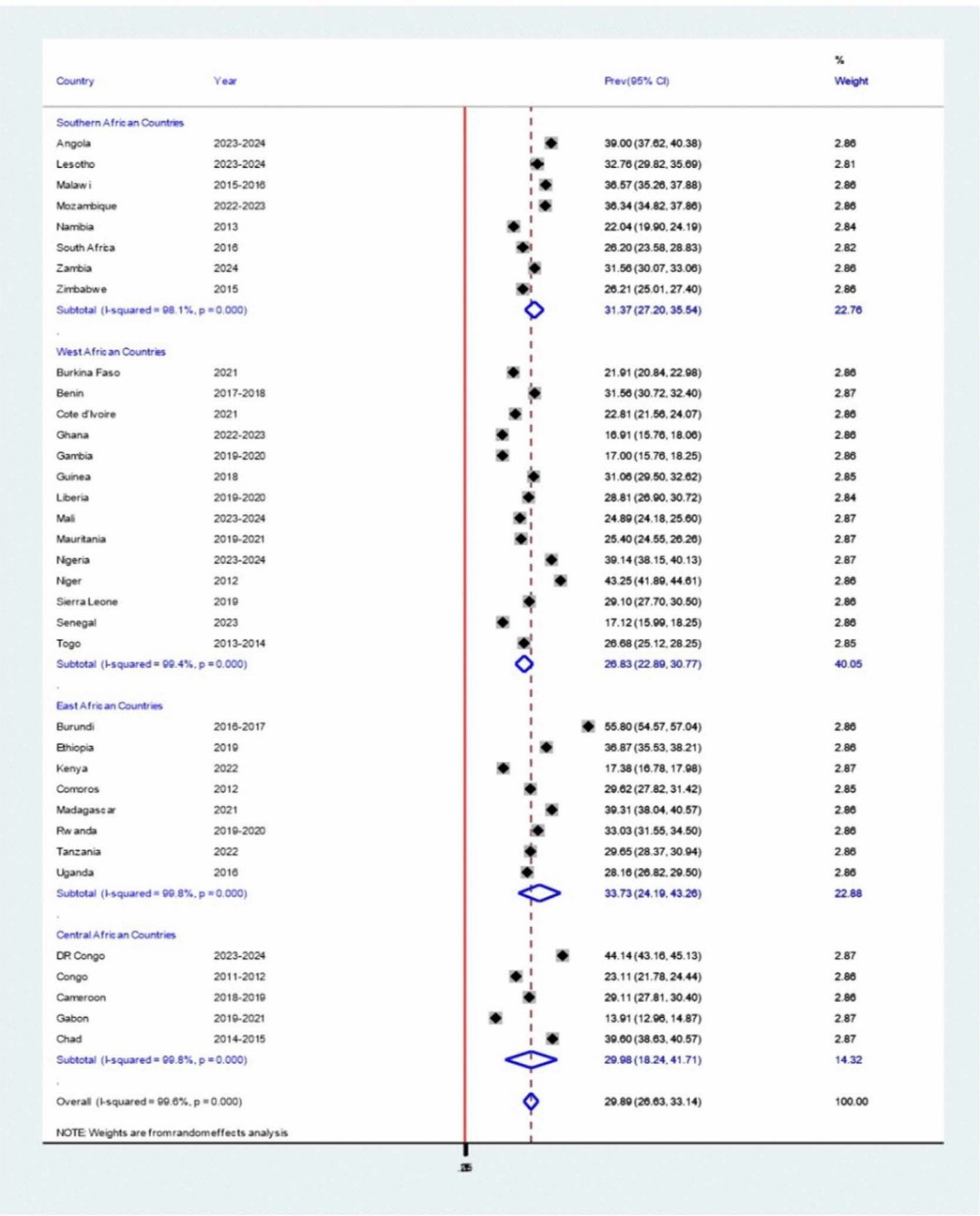

**Fig 4. Forest plot showing the pooled prevalence of stunting among children under 5 years of age in sub-Saharan Africa, by subregion, based on a random-effects model analysis, 2011-2024.**

of stunting was 2.07 times higher among those aged 12–23 months (aPR = 2.07, 95% CI: 2.00–2.14). It was 2.41 times higher among children aged 24–35 months (aPR = 2.41, 95% CI: 2.32–2.50), 2.32 times higher among those aged 36–47 months (aPR = 2.32, 95% CI: 2.23–2.42), and 1.81 times higher among children aged 48–59 months (aPR = 1.81, 95% CI: 1.71–1.90) (Table 5).

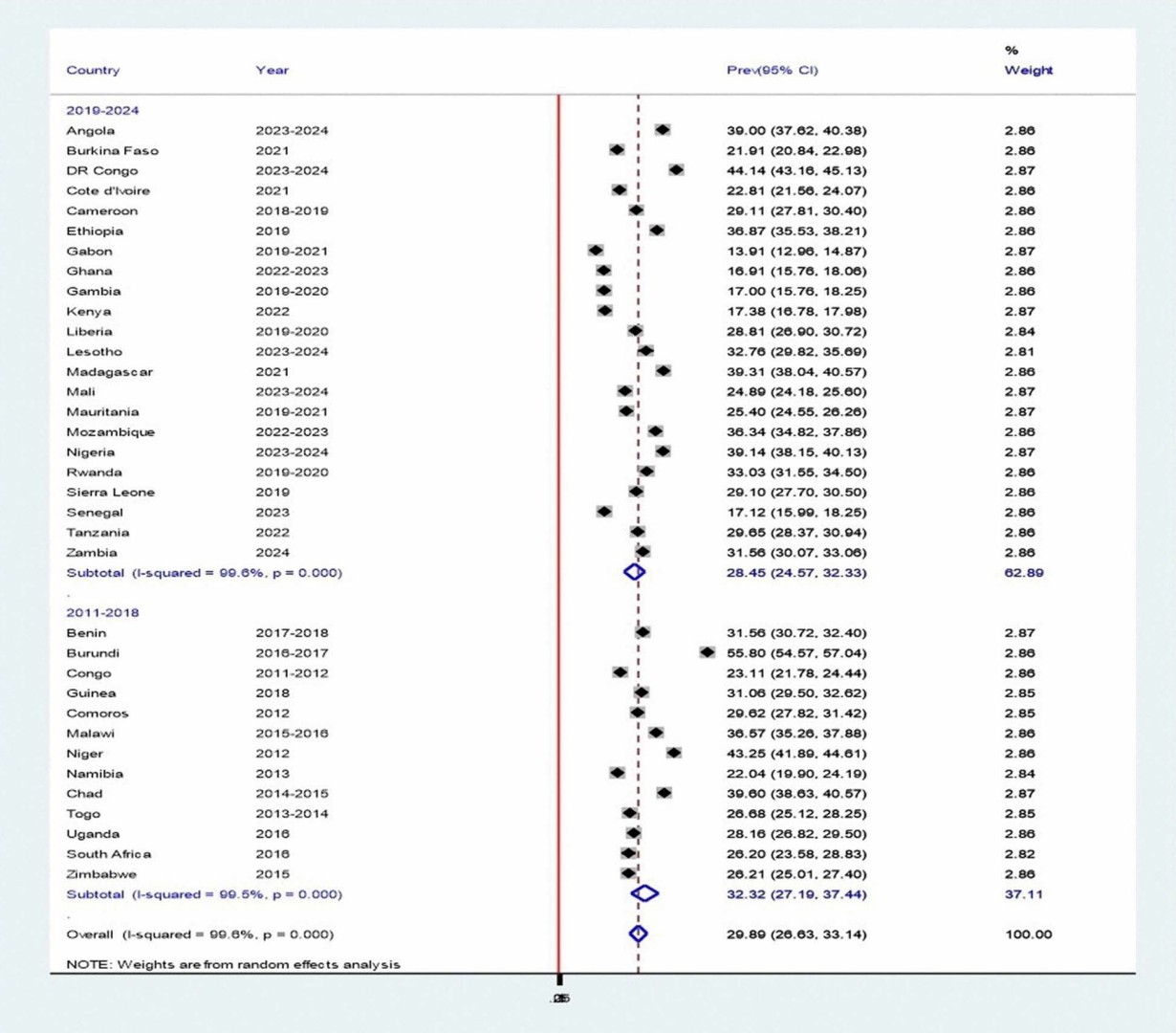

**Fig 5. Forest plot showing the pooled prevalence of stunting among children under 5 years of age in sub-Saharan Africa, by survey year category, based on a random-effects model analysis, 2011-2024.**

**Table 3. Meta-regression for the studies of pooled stunting among children under 5 years of age in 35 sub-Sharan Africa, 2011-2024.**

| Outcome categories | Variables | Coefficients | P-value | [95% conf. interval] |
|---|---|---|---|---|
| *Pooled stunting among under five children* | Year of Survey | 0.99 | 0.68 | 0.96, 1.02 |
| | Sample size | 1.00 | 0.65 | 0.99, 1.00 |
| | Countries | 0.99 | 0.22 | 0.98, 1.00 |
| | Sub-region | 1.06 | 0.19 | 0.96, 1.17 |

Children born in the fifth or higher birth order had an 8% higher prevalence of stunting compared to those born as first-order children (aPR = 1.08, 95% CI: 1.03–1.12). Using children perceived as large at birth as the reference category, the prevalence of stunting was 16% higher among those perceived as average size at birth (aPR = 1.16, 95% CI:

**Table 4. Selection of most parsimonious model (measures of variation) at cluster or community level for stunting among children under 5 years of age in 35 SSA countries, 2011–2024.**

| Measure of variation | Models | | | | |
|---|---|---|---|---|---|
| | Null-Model | Model I | Model II | Model III | Model IV |
| Cluster-level variance (SE) | 0.5(0.05) | 0.15(0.001) | 0.2(0.03) | 0.3(0.04) | 0.16(0.025) |
| Intra-class correlation (%) or VPC | 26.8% | 13.8% | 12.7% | 11.4% | 10.2% |
| Explained variance (%) or PCV | *Reference* | 70% | 60% | 40% | 47.2% |
| Median odds ratio | 1.96 | 1.44 | 1.5 | 1.68 | 1.46 |
| Model summary | | | | | |
| Akaike's information criteria | 245122.6 | 138191.7 | 169678.7 | 241377.5 | **98254.67** |
| Bayesian information criteria | 245153.1 | 138453.6 | 169807 | 241448.8 | **98639.09** |
| Log-likelihood | −122558.28 | −69068.872 | −84826.365 | −120681.75 | **−49086.333** |
| Deviance | 245116.55 | 138137.74 | 169652.73 | 241363.5 | **98172.666** |

*VPC = Variance Partitioning Coefficient, PCV = Proportion Change in Variance*

1.13–1.19) and 48% higher among those perceived as small size at birth (aPR = 1.48, 95% CI: 1.43–1.52). Compared to children whose mothers had a higher level of formal education, the prevalence of stunting was 1.94 times higher among children whose mothers had no formal education (aPR = 1.94, 95% CI: 1.74–2.18), 1.95 times higher among those whose mothers had only primary education (aPR = 1.95, 95% CI: 1.74–2.19), and 1.68 times higher among those whose mothers had secondary education (aPR = 1.68, 95% CI: 1.51–1.88).

Using mothers aged 25 years and above as the reference category, the prevalence of stunting was 9% higher among children of mothers aged 15–24 years (aPR = 1.09, 95% CI: 1.06–1.13). Compared to children whose mothers had four or more antenatal care (ANC) visits, the prevalence of stunting was 17% higher among those whose mothers had no ANC visits (aPR = 1.17, 95% CI: 1.13–1.21) and 11% higher among those whose mothers had only one to three visits (aPR = 1.11, 95% CI: 1.09–1.14). In addition, children delivered at home had an 11% higher prevalence of stunting than those delivered in health facilities (aPR = 1.11, 95% CI: 1.08–1.14). Children whose mothers did not attend postnatal visits had a 3% higher prevalence of stunting compared with those whose mothers attended such visits (aPR = 1.03, 95% CI: 1.01–1.05). Moreover, the prevalence of stunting was 1.04 times higher among children of single mothers than among those of married mothers (aPR = 1.04, 95% CI: 1.02–1.07).

This study revealed that maternal BMI was significantly associated with childhood stunting. Compared to children of mothers with a normal BMI, the prevalence of stunting was 12% higher among children of underweight mothers (aPR = 1.12, 95% CI: 1.08–1.16). In contrast, the prevalence of stunting was 19% lower among children of overweight mothers compared to those of mothers with a normal BMI (aPR = 0.81, 95% CI: 0.77–0.84). Similarly, the prevalence of stunting was 12% lower among children of obese mothers compared to those of mothers with a normal BMI (aPR = 0.88, 95% CI: 0.85–0.90).

This study revealed that household size was significantly associated with childhood stunting. Compared to children from smaller households, the prevalence of stunting was 4% higher among children from large households (aPR = 1.04, 95% CI: 1.04–1.08). In addition, children whose mothers reported no exposure to mass media such as television, radio, or newspapers had an 11% higher prevalence of stunting compared to children whose mothers had some media exposure (aPR = 1.11, 95% CI: 1.08–1.13).

Moreover, compared to children from rich households, the prevalence of stunting was 31% higher among children from poor households (aPR = 1.31, 95% CI: 1.24–1.38). Similarly, the prevalence of stunting was 23% higher among children from middle wealth index households compared to those from rich households (aPR = 1.23, 95% CI: 1.17–1.29).

**Table 5. Bivariable and multivariable multilevel mixed-effects robust Poisson regression analysis of determinants of stunting among children under 5 years of age in 35 SSA countries, 2011–2024.**

| Covariates | Stunting | | cPR with 95% | aPR with 95% |
|---|---|---|---|---|
| | Yes | No | | |
| **Sex of the child** | | | | |
| Male | 31,604(32.61) | 65,303(67.39) | 1.16(1.14, 1.18) * | *1.24(1.21, 1.26) ** |
| Female | 26,642(28.03) | 68,402(71.97) | 1 | 1 |
| **Age of the child (months)** | | | | |
| Less than 12 months | 7,156(16.96) | 35,029(83.04) | 1 | 1 |
| 12–23 months | 13,492(33.81) | 26,415(66.19) | 1.99(1.93, 2.06) * | *2.07(2.00, 2.14) ** |
| 24–35 months | 14,225(38.49) | 22,736(61.51) | 2.26(2.19, 2.34) * | *2.41(2.32, 2.50) ** |
| 36–47 months | 13,108(34.76) | 24,599(65.24) | 2.05(1.98, 2.12) * | *2.32(2.23, 2.42) ** |
| 48–59 months | 10,264(29.17) | 24,926(70.83) | 1.72(1.66, 1.78) * | *1.81(1.71, 1.90) ** |
| **Birth order** | | | | |
| First | 11,640(27.88) | 30,115(72.12) | 1 | 1 |
| 2–4 | 27,615(29.12) | 67,224(70.88) | 1.04(1.01, 1.06) * | 1.01(0.98, 1.05) |
| 5 or higher | 18,991(34.31) | 36,367(65.69) | 1.21(1.18, 1.24) * | *1.08(1.03, 1.12) ** |
| **Perceived size at birth** | | | | |
| Large | 15,044(28.13) | 38,434(71.87) | 1 | 1 |
| Average | 25,391(30.97) | 56,608(69.03) | 1.10(1.07, 1.12) * | *1.16(1.13, 1.19) ** |
| Small | 10,537(38.71) | 16,682(61.29) | 1.37(1.34, 1.41) * | *1.48(1.43, 1.52) ** |
| **Mothers' educational status** | | | | |
| No education | 25,867(36.38) | 45,241(63.62) | 3.84(3.49, 4.22) * | *1.94(1.74, 2.18) ** |
| Primary | 20,175(33.07) | 40,839(66.93) | 3.49(3.17, 3.84) * | *1.95(1.74, 2.19) ** |
| Secondary | 11,385(22.32) | 39,619(77.68) | 2.37(2.15, 2.61) * | *1.68(1.51, 1.88) ** |
| Higher | 818(9.28) | 8,006(90.72) | 1 | 1 |
| **Mother's age (years)** | | | | |
| 15-24 | 16,514(31.63) | 35,694(68.37) | 1.05(1.03, 1.07) * | *1.09(1.06, 1.13) ** |
| 25-49 | 41,732(29.86) | 98,012(70.14) | 1 | 1 |
| **Number of ANC visits** | | | | |
| No visit | 4,919(40.36) | 7,269(59.64) | 1.36(1.32, 1.41) * | *1.17(1.13, 1.21) ** |
| 1 to 3 visits | 12,762(32.19) | 26,884(67.81) | 1.10(1.08, 1.12) * | *1.11(1.09, 1.14) ** |
| 4 and above visit | 40,565(28.95) | 99,552(71.05) | 1 | 1 |
| **Place of delivery** | | | | |
| Home | 16,649(38.46) | 32,199(26.68) | 1.44(1.41, 1.47) * | *1.11(1.08, 1.14) ** |
| Health facility | 38,186(61.54) | 88,466(73.32) | 1 | 1 |
| **Postnatal care visits** | | | | |
| Yes | 11,787(26.33) | 32,986(73.67) | 1 | 1 |
| No | 25,394(30.15) | 58,832(69.85) | 1.14(1.11, 1.17) * | *1.03(1.07, 1.05) ** |
| **Marital status** | | | | |
| Single | 12,499(29.20) | 30,311(70.80) | 0.94(0.92, 0.96) * | *1.04(1.02, 1.07) ** |
| Married | 45,747(30.67) | 103,39(69.33) | 1 | 1 |
| **Mothers age at 1st birth** | | | | |
| Less than 20 | 36,334(32.84) | 74,309(67.16) | 1.20(1.18, 1.22) * | 1(0.97, 1.02) |
| 20 and above | 21,912(26.95) | 59,397(73.05) | 1 | 1 |
| **Maternal nutritional status** | | | | |
| Underweight | 5,131(40.76) | 7,459(59.24) | 1.19(1.15, 1.22) * | *1.12(1.08, 1.16) ** |

*(Continued)*

 

**Table 5.** (Continued)

| Covariates | Stunting | | cPR with 95% | aPR with 95% |
|---|---|---|---|---|
| | **Yes** | **No** | | |
| Normal weight | 31,342(34.21) | 60,270(65.79) | 1 | 1 |
| Overweight | 6,036(23.08) | 20,113(76.92) | 0.68(0.66 0.70) * | *0.81(0.77, 0.84)* ** |
| Obese | 15,736(25.55) | 45,863(74.45) | 0.75(0.73, 0.77) * | *0.88(0.85, 0.90)* ** |
| **Household size** | | | | |
| Small | 4,882(29.30) | 11,782(70.70) | 1 | 1 |
| Medium | 3,719(29.23) | 9,004(70.77) | 1(0.95, 1.04) | 0.99(0.94, 1.05) |
| Large | 49,644(30.54) | 112,919(69.46) | 1.04(1.01, 1.07) * | *1.04(1.04, 1.08)* ** |
| **Mass media exposure** | | | | |
| No | 25,077(38.34) | 40,337(61.66) | 1.44(1.41, 1.47) * | *1.11(1.08, 1.13)* ** |
| Yes | 33,169(26.21) | 93,368(73.79) | 1 | 1 |
| **Wealth index** | | | | |
| Poor | 16,637(38.34) | 52,993(39.63) | 2.26(2.17, 2.36) * | *1.31(1.24, 1.38)* ** |
| Middle | 11,958(31.10) | 26,492(19.81) | 1.83(1.75, 1.91) * | *1.23(1.17, 1.29)* ** |
| Richer | 9,793(26.42) | 54,221(40.55) | *1* | *1* |
| **Place of residence** | | | | |
| Urban | 14,229(21.57) | 51,754(78.43) | 1 | 1 |
| Rural | 44,017(34.94) | 81,952(65.06) | 1.61(1.56, 1.66) * | *1.07(1.04, 1.11)* ** |

*cPR*: Crude Prevalence Ratio, *aPR*: Adjusted Prevalence Ratio, *: Significant in crude prevalence ratio, **: Significant in adjusted prevalence ratio

Furthermore, children living in rural areas had a 7% higher prevalence of stunting compared to their urban counterparts (aPR = 1.07, 95% CI: 1.04–1.11).

## Discussion

Reducing stunting in sub-Saharan Africa (SSA) continues to pose a significant challenge, yet various programmatic initiatives are being introduced to drive progress. The World Bank's "All Hands-on Deck" initiative advocates for a multisectoral strategy, involving sectors such as agriculture, education, social protection, and water, sanitation, and hygiene (WASH) [41]. Similarly, UNICEF's programs prioritize maternal, infant, and young child nutrition, micronutrient supplementation, the management of severe acute malnutrition, nutrition in emergencies, and enhancing water and sanitation at the community level [42]. Despite these efforts, the burden of stunting remains high in sub-Saharan African countries. These integrated approaches are specifically designed to accelerate stunting reduction in SSA, with a particular emphasis on East Africa, where the challenge persists [4].

This study analyzed pooled data from nationally representative surveys conducted across 35 sub-Saharan African (SSA) countries between 2011 and 2024. The findings revealed that the overall pooled prevalence of stunting among children under 5 years of age in SSA was 29.89% (95% CI: 26.63, 33.14%). The lowest stunting prevalence rate was recorded in Gabon at 13.74%, while the highest prevalence of stunting was observed in Burundi at 55.74%. Among the 35 countries analyzed, 14 had a stunting prevalence of 30% or higher among children under 5 years of age. This finding is consistent with World Health Organization prevalence threshold or cut-off values for public health significance in which prevalence of stunting 20 to <30% is high and greater than or equal to 30% is very high [43]. Additionally, the pooled prevalence of stunting aligns with findings from studies carried out in 65 Low- and Middle-Income Countries(29.0%) [44], 62 LMICs worldwide(29.1%) [45], Tajikistan (27%) [46], and Tanzania (28%) [47].

However, the observed prevalence is lower than that reported in previous studies conducted in Ethiopia (33.58%) [48], Low- and Middle-Income Countries(38.8%) [49], East Africa(33.3%, 39%) [50,51], Sub-Saharan Africa(33.2%) [51], Pakistan(44%) [52], Ethiopia(37%) [53], Kenya (39%) [54], Sub-Saharan Africa(35%) [4], Tanzania(35%) [55], Vietnam (44%) [56], Chitwan (37.3%) [57], and Belahara VDC (37%) [58],. Additionally, this finding was higher than the prevalence of stunting reported in studies conducted in China (4.4%) [59], Bangladesh (25%) [60], Bhutan(21%) [61], and UNICEF-WHO-WB Joint Child Malnutrition Estimates(22.3%) [62].

A potential explanation for this finding could be the differences in nutritional practices, sociocultural and environmental factors across the countries. Variations in the scope of the studies, the year of research, population characteristics, and methodologies employed might have contributed to the observed differences. For instance, earlier studies were limited in scope, focusing on individual countries and utilizing diverse analytical approaches [4,24,63–66] and some studies also included a restricted number of countries and population characteristics such as ages [4]. Moreover, another main reason could be attributed to the prolonged malnutrition among children under 5 years of age in Sub-Saharan Africa (SSA), stemming from persistent poverty in the region [67]. Furthermore, the limited economic resources of many African nations can hinder access to sufficient and nutritious food, contributing to the prevalence of stunting [68].

Although the pooled prevalence of stunting among children under 5 years of age in 35 SSA countries was estimated at 29.89% (95% CI: 26.63, 33.14%), the exceptional high heterogeneity ($I^2 = 99.6\%$, $p < 0.000$) underscores substantial contextual variation across countries. This variation is not statistical only but also reflects meaningful differences in underlying determinants, health system capacity, and sociopolitical environments. Subgroup analysis revealed that East African countries had highest stunting burden (33.73%), while West African showed comparatively lower prevalence (26.83%), suggesting region drivers such as climate vulnerability, food system resilience, or maternal education levels.

However, meta-regression indicated that broad variables like survey year, sample size, country, and sub-region did not significantly explain this heterogeneity (all p-values > 0.05), pointing to the influence of more unmeasured factors such as governance quality, conflict exposure, or local nutrition policies. These findings highlight the limitations of the relying solely on pooled estimate for policy guidance and emphasizes the need for country level diagnostics to identify context specific risk factors. Future research should explore these modifiers systematically to inform tailored interventions rather than one size fits all strategies.

Our findings revealed that stunting among children under 5 years of age in sub-Saharan African countries is shaped by a mix of individual, household-level, and contextual factors. One significant predictor was the sex of the child, with male children more likely to experience stunting than females. This finding aligns with a study conducted in Sub-Saharan African (SSA) countries, as well as studies conducted in Zambia and Ethiopia [4,65,69,70]. The consistency of this association across diverse settings suggests a robust pattern that may reflect biological vulnerability, differential care practices, or gendered exposure to environmental risks. Evidence from longitudinal studies in sub-Saharan Africa indicates that male children are more prone to growth faltering during the first year of life [71]. However, findings from certain interventions program such as nutritional supplementation and maternal education initiatives suggest that these sex-based disparities in stunting may diminish over time [72].

This indicates that while male children may be biologically predisposed to stunting, contextual factors such as household factors, caregiving practices, and health system responsiveness can modify these risks. What remain uncertain is whether the sex differential is primarily driven by biological susceptibility or by modifiable social determinants. Future research, particularly longitudinal interventions studies, is needed to clarify and separate the overlapping biological and socio-behavioral pathways, in order to determine whether targeted strategies for boys could achieve greater reductions in stunting prevalence.

The prevalence of stunting was significantly higher among older children in the age groups of 12–23 months, 24–35 months, 36–47 months, and 48–59 months compared to children under 12 months of age. This finding indicates that the risk of stunting rises as a child grows older. Similar age-related patterns have been reported in studies conducted in

Ethiopia [48,53,65,73], Somalia [74], Rwanda [75], and Zimbabwe [76], suggesting that growth faltering often becomes more pronounced after infancy.

A prospective birth cohort studies have demonstrated that growth deficit often accelerates between 6 and 24 months, reflecting both biological vulnerability and the transition to inadequate diets and increased exposure to infections [77,78]. Interventions trial have further shown that timely interventions during this critical window period can reduce stunting prevalence [79]. The novel contribution of this study is the documentation of this established association within our specific, previously less investigated population, thereby strengthening the geographical evidence base. What remain uncertain, however is, the extent to which age-related differences are modifiable through targeted interventions and whether the persistency of stunting in to later childhood reflects irreversible early childhood damage or ongoing exposure to adverse conditions.

Children born in the fifth or higher birth order had an 8% higher prevalence of stunting compared to those born as first-order children. This finding aligns with earlier studies that have examined child nutritional status, considering birth order as a significant confounding factor [80–83]. Cohort studies have shown that later born children often experience reduced access to household resources, shorter breastfeeding duration, and greater exposure to infections, which collectively contribute to growth faltering [84,85]. Interventions study also demonstrated that empowering mothers to space births and manage household resources can mitigate the nutritional advantages faced by higher order children [86,87]. However, the extent to which birth order effects are biologically driven versus socially mediated, and whether targeted interventions such as birth spacing programs or nutrition support for larger family can fully offset these risk remains uncertain.

The perceived size of a baby at birth showed a significant association with stunting status, with children born small and average size more likely to be stunted compared to those born large. This finding is consistent with studies conducted in Africa and other developing countries, such as Indonesia and Ethiopia [88–90] as well as Nepal [91], Bangladesh [92] and Ethiopia [93]. Cohort studies conducted in Ethiopia and Malawi have shown that children born small remain at elevated risk of stunting well into the first two years of life, even after accounting for postnatal feeding practices [94]. Another interventional study on maternal nutrition supplementation programs in Ghana and Tanzania, further confirmed that improving maternal diet and antenatal care can reduce the incidence of the low birth weight and subsequently lower stunting prevalence [95,96]. However, uncertainties remain regarding the extent to which postnatal interventions (e.g., complementary feeding programs) can fully offset the disadvantage of being born small, suggesting that prevention strategies must prioritize maternal health and antenatal care.

The study revealed that compared to children whose mothers had a higher level of formal education, the prevalence of stunting was 1.94 times higher among children whose mothers had no formal education, 1.95 times higher among those whose mothers had only primary education, and 1.68 times higher among those whose mothers had secondary education. This finding is supported by a systematic review of studies published worldwide between 2004 and 2014, as well as by research conducted in Ethiopia, Zambia and Nepal [4,97–100]. Previous intervention studies have established that maternal education is a protective factor. For example, community based nutritional programs in Uganda and Tanzania show that incorporating maternal literacy and health educations components resulted in significantly lower stunting prevalence in intervention group versus control group [101,102]. However, whether maternal nutritional counselling can only compensate for the disadvantage of low maternal education in resource-constrained settings remains uncertain. Future research should explore whether combining educational empowerment with direct nutrition interventions yields synergistic effects in reducing stunting.

Children born to mothers aged 15–24 were more likely to experience higher levels of stunting compared to those born to mothers aged 25–49. This finding is consistent with the study conducted in Ethiopia [103]. This may be due to the fact that babies born to younger mothers are more prone to preterm birth and low birth weight, which can increase the risk of neonatal infections and malnutrition, including stunting [104]. Another study also showed that younger maternal age is

often associated with limited caregiving experience, reduced autonomy in household decision-making, and lower access to health and nutrition services contributing higher stunting prevalence [105]. However, further research needs maternal age itself to act as an independent risk factor versus being mediated by socioeconomic status, maternal education or health service utilization.

Mothers' antenatal care (ANC) visits were strongly associated with child stunting. Children whose mothers did not attend any antenatal follow-up and 1–3 ANC visits more likely to be stunted compared to those whose mothers had four or more ANC visits. This finding is supported by a study conducted in Ethiopia [106–108], Zambia [109], Latin America [110,111], and Bangladesh [112]. This may be attributed to the fact that antenatal care (ANC) programs are structured to identify high-risk mothers and deliver nutritional and educational support, such as advice on food hygiene, diet, and lifestyle. These initiatives target key factors that play a significant role in enhancing child nutrition. ANC programs are advocated as an effective approach to lowering the prevalence of low birth weight, and evidence highlighting their effectiveness in reducing adverse pregnancy outcomes in developing countries is steadily increasing [113,114]. However, barriers such as inequitable health system coverage, geographical isolation and weak governance in service delivery restrict women access to timely and quality antenatal care services [115,116].

In this study, the place of delivery showed a significant association with child stunting. Specifically, children born at home were more likely to be stunted compared to those born at health facilities. This finding aligns with a study conducted in Ethiopia [4,117]. A possible explanation for this association is that mothers who deliver at home are less likely to receive proper postnatal care, including vaccinations, which are critical for the healthy growth and development of children. Vaccinations help prevent several vaccine-preventable diseases and ensure children receive essential nutrients, contributing to better overall health outcomes [118].

The study revealed that postnatal care visits were associated with childhood stunting. Mothers who did not attend postnatal care services were 1.03 times more likely to have a stunted child compared to those who did. This finding is consistent with a study conducted in Ethiopia [119]. Study confirmed that strengthening maternal contact with health systems during the postnatal period contributes to improved child growth outcomes [120]. However, limited attendance at postnatal care is not solely a matter of maternal choice but reflects broader structural determinants. Barriers such as poverty, inequitable health system coverage, weak governance in service delivery and in some context conflict or geographical isolation restrict women access to timely and quality postnatal services [115,116].

Addressing stunting therefore requires structural interventions such as equitable health infrastructure, improving governance and accountability in maternal and child health programs, and strengthening social protection alongside efforts to promote household-level health-seeking behaviors. Additionally, the extent to which postnatal care independently reduces stunting versus acting synergistically with other maternal and child health interventions such as antenatal care, breastfeeding promotion, and community nutrition programs remains uncertain.

Another factor associated with stunting was marital status. In the study, the prevalence of child stunting was higher among children of single mothers than among children with married mothers. This finding consistent with previous study [121]. The possible explanation may be attributed to socioeconomic challenges, such as limited financial resources, reduced access to healthcare, and lower social support. Single mothers often face greater difficulties in providing adequate nutrition and care for their children due to these constraints. Additionally, the stress and time demands associated with single parenthood may further hinder their ability to ensure optimal child-rearing conditions, contributing to poorer growth outcomes. These factors collectively increase the risk of stunting in children of single mothers [122].

This study found that maternal body mass index was associated with childhood stunting. Compared to children whose mothers had a normal body mass index (BMI), the prevalence of stunting was higher among children of underweight mothers and lower among those of overweight mothers, and obese mothers. These findings were consistent with previous study conducted in low-income and middle-income countries [123], Ethiopia [124,125], Bangladesh [126] and Nigeria [6].

While poor maternal nutrition likely contributes to inadequate nutrient transfer during pregnancy and lactation, resulting in impaired child growth, the relationship between maternal BMI and stunting extends beyond individual level behaviors. Lower stunting rates among children of overweight or obese mothers may reflect not only higher caloric intake but also broader socioeconomic advantages, such as improved household food security, access to diverse diets, and greater resilience to shocks [127].

These maternal outcomes are shaped by structural determinants, including poverty, food system functioning, governance and conflict that influence both maternal nutrition and child health. For instance, household insecure regions or conflict affected setting may face systematic barriers to adequate nutrition regardless of maternal practices [127,128]. Thus, while balanced maternal nutrition remains critical, these findings underscore the needs to situate maternal and household factors within wider structural contexts. Addressing stunting requires not only promoting individual behavior change but also tackling systematic inequities through poverty reduction, strengthening food systems, improving governance and ensuring stability.

This study revealed that child stunting was high among households with large family sizes. This finding is consistent with a previous study conducted in sub-Saharan Africa [129], and Ethiopia [130,131]. This may be due to the fact that higher prevalence of child stunting in households with large family sizes may be due to resource dilution, where limited financial and food resources are spread thinly among many family members, reducing the quantity and quality of nutrition available for each child. Larger families may also face challenges in providing adequate healthcare and sanitation, further exacerbating the risk of stunting. Additionally, caregivers in large families may have less time and capacity to ensure proper feeding practices and childcare, contributing to poorer growth outcomes [132].

The results of this study also revealed that mothers' exposure to mass media was significantly associated with high prevalence of child stunting. Children whose mothers had not media exposure were more likely to be stunted compared to those whose mothers had exposure to media. This finding was consistent with studies conducted in Pakistan, Bangladesh and Ethiopia [4,133,134]. Media plays a vital role in promoting sociocultural and economic development, which can contribute to improved nutritional outcomes [135]. However, access to mass media is shaped by systemic issues such as poverty, inequitable information and communication infrastructures, governance of public health messaging, and in some contexts, conflict that limits dissemination of reliable information [136]. These structural barriers constrain households' ability to access knowledge about nutrition, sanitation, and child health, thereby reinforcing cycles of stunting [137]. Therefore, addressing these challenges require system level interventions.

Our study revealed a significant positive association between the household wealth index and child stunting. Compared to children from the richest household wealth index, child from the poor and middle households' wealth index had higher odds of being stunted. While these findings align with a systematic review and meta-analysis conducted in Africa between 2000 and 2013 and study conducted in Sub-Saharan Africa [4,138], as well as studies from low-income countries such as Nepal, Zambia, and Nigeria [99,100,139], the implications extend beyond household-level economic status [140]. Household wealth reflects broader structural determinants including entrenched poverty, inequitable food systems, weak governance, and in some settings, the destabilizing effects of conflict that constrain access to adequate nutrition, health services, and safe environments for children [141,142]. Thus, stunting should be understood not only as a consequence of limited household resources but also as a manifestation of systemic inequities.

Furthermore, contextual-level variables specifically, place of residence were significantly associated with child stunting. Children from rural areas had higher prevalence of being stunted compared to their urban counterparts. A finding supported by studies conducted in Pakistan and Ethiopia [143,144]. This may be attributed to the better nutritional status of urban children, which stems from improved maternal prenatal and postnatal care, higher-quality complementary feeding practices, and better immunization coverage. Additionally, urban children benefit from advantages such as family employment opportunities, stronger social and family networks, and greater access to healthcare and social services, all of which contribute to their improved nutritional status compared to rural children [145].

**Strength and limitation of the study**

This study utilized a recent, nationally representative dataset encompassing 191,953 children under five from 35 sub-Saharan African countries surveyed between 2011 and 2024. The large sample size substantially enhanced the statistical power to detect true effect sizes and improved the generatability of the findings across diverse settings. Advanced statistical methods, including multilevel mixed-effects Poisson regression with robust variance, were employed to strengthen the analysis and account for the hierarchical nature of the DHS data. By pooling data across countries, the study estimated the prevalence of stunting and its determinants, revealing significant and concerning heterogeneity between countries that persisted even after subgroup and sensitivity analyses. Importantly, the inclusion of key maternal and child health variables such as antenatal care (ANC) and postnatal care (PNC) visits, and maternal body mass index (BMI) addressed gaps overlooked in previous studies [4], thereby providing a more comprehensive understanding of stunting determinants.

Nonetheless, several limitations should be acknowledged. Health system variables, including health insurance coverage, the availability and type of health facilities, and broader country-level contextual factors such as political stability, were not assessed. Furthermore, the cross-sectional design of the DHS surveys precludes establishing causal relationships between independent and dependent variables. While significant associations were identified, determinants such as maternal nutritional status and household media exposure are likely influenced by broader socioeconomic conditions that also directly affect child stunting. Maternal BMI, for example, may shape child nutrition but is itself conditioned by household food insecurity and poverty, while media exposure may influence health behaviors yet remains dependent on socioeconomic access. Without longitudinal data, the temporal of these relationships remains uncertain. Therefore, the findings should be interpreted as highlighting important populations-level correlates rather than confirming causal pathways. Future prospective cohort studies are warranted to disentangle these relationships and clarify the causal mechanisms underlying child stunting in SSA.

## Conclusion

In conclusion, this study analyzed pooled data from 35 sub-Saharan African (SSA) countries and found an overall stunting prevalence of 29.72% among children under 5 years of age, with significant variations across countries, ranging from 13.74% in Gabon to 55.74% in Burundi. Fourteen countries had stunting rates exceeding 30%, indicating a very high public health concern as per WHO thresholds.

Our study identified individual, household, and contextual factors significantly linked to stunting in children under 5 years of age. At the individual level, key predictors included being male, increased age of child, higher birth order (fifth or more), small or medium birth size, insufficient antenatal care (ANC) visits (none or 1–3), lack of postnatal visits, lower maternal education (no formal, primary, or secondary education), home delivery, single, unmarried status, and maternal underweight, all of which were associated with increased prevalence of stunting. In contrast, maternal overweight or obesity were linked to reduced prevalence of stunting. At the household level, factors such as lack of maternal media exposure (television, radio, or newspapers), large family size, and lower household wealth (poor, middle, rich) were associated with higher prevalence of stunting. Additionally, at the contextual level, being rural residence was a significant predictor of stunting.

To address persistently high prevalence of stunting in sub-Saharan Africa, interventions should be clearly prioritized by feasibility and timeline but framed within both household level and structural determinants. Immediate actions remain critical such as improving maternal and child healthcare access through increased antenatal and postnatal care visit, promoting facility-based deliveries, and enhancing maternal education to strengthen feeding practices and nutrition awareness. Expanding media outreach can also provide rapid gains in health and nutrition knowledge.

However, this effort must be complemented by long-term structural and systems level investments that tackle the root cause of the stunting. Poverty alleviation, resilient food systems, and equitable access to nutritious diets are central to

reducing vulnerability. Governance and political stability play a decisive role in ensuring effective implementation of nutrition policies while conflict and displacement exacerbate food insecurity and health service disruptions. Addressing gender inequality in child nutrition, improved access to clean water and sanitation, expanding clean cooking fuels, and bridging between urban-rural disparities through strengthened healthcare infrastructure and social protection programs are equally vital.

A multisectoral approach like integrating health, education, agriculture, food systems and WASH sectors is essential to ensure both short term impact and sustainable improvements in child growth and developments. Future research should incorporate health systems variables like facility availability and quality alongside country level factors like governance, policy coherence and conflict dynamic. Longitudinal design is needed to establish casual pathways and guide more effective policy strategies that balance individual behavior change with structural reforms.

## Acknowledgments

The authors thank ICF International for granting access to the SSA DHS data set used in this study.

## Author contributions

**Conceptualization:** Abdulkerim Hassen Moloro, Kusse Urmale Mare.

**Data curation:** Abdulkerim Hassen Moloro.

**Formal analysis:** Abdulkerim Hassen Moloro, Kebede Gemeda Sabo, Kusse Urmale Mare, Oumer Abdulkadir Ebrahim, Begetayinoral Kussia Lahole.

**Methodology:** Abdulkerim Hassen Moloro, Oumer Abdulkadir Ebrahim, Begetayinoral Kussia Lahole.

**Software:** Abdulkerim Hassen Moloro, Kebede Gemeda Sabo.

**Validation:** Abdulkerim Hassen Moloro.

**Writing – original draft:** Abdulkerim Hassen Moloro, Kusse Urmale Mare.

**Writing – review & editing:** Abdulkerim Hassen Moloro, Kebede Gemeda Sabo, Kusse Urmale Mare, Beriso Furo Wengoro, Eshetu Elfios Endrias, Roda Mehadi Ibrahim, Teshager Dubie, Oumer Abdulkadir Ebrahim, Begetayinoral Kussia Lahole.

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
