## [Decision Letter · Decision Letter 0]

3 Nov 2025

Dear Dr. Moloro,

Thank you for submitting your manuscript to PLOS ONE. After careful consideration, we feel that it has merit but does not fully meet PLOS ONE’s publication criteria as it currently stands. Therefore, we invite you to submit a revised version of the manuscript that addresses the points raised during the review process.

You did not line-number your manuscript to help reviewers pinpoint the exact areas requiring revision in the manuscript.There is inconsistent use of terms for your outcome variable: "stunting among U-5 children", "stunting", "childhood stunting", "stunting among under five children",Abstract section:Your abstract exceeds 300 words, so you need to be parsimonious in the reporting of your findings. You can report the most salient findings: **the child's sex, age & birth size; the mother's education; and household wealth,** i.e, those with aPR ≥±20%Please revise this statement "*Key variables such as antenatal care visits, postnatal care, and maternal nutritional indicators which previous study not accounted incorporated in to analysis* ". **Suggestion** : *Key variables such as antenatal care visits, postnatal care, and maternal nutritional indicators, which previous studies did not account for, are incorporated into the analysis* .Please revise this statement: "*The adjusted prevalence ratio and 95% confidence intervals were used to indicate statistical significance and the strength of associations*". **Suggestion** : "*The adjusted prevalence ratios and their 95% confidence intervals were used to assess the strength and statistical significance of associations*".The abstract results are overloaded with too much information. Kindly reduce the results to the salient ones as suggested above. In fact, the entire multivariate analysis results in the result section need to be rewritten. Again, the term "children older age" is ambiguous, and the aPR value of 2.32, which you stated, is misleading. It is better to say: "**being aged 12 months or older" (aPR: ≥1.81, p<0.01)** . Also, "mother without a higher education **(aPR: ≥1.68, p<0.01), and living in a poor or average household (aPR: ≥1.23, p<0.01).**Method sectionI suggest you merge the poorest and the poorer categories into one category, "poor", and merge the richer and the richest into one category as "rich", thus you will have three categories for the wealth index. This will provide a much better explanation of your findings.The 2nd paragraph on Page 8: I see that you included "source of drinking water", "type of toilet facility", and "type of cooking" in your independent variable list. I suggest you exclude these variables because they have already been used to estimate the household wealth index (composite index). Using them again with the household wealth index in the model may not be statistically appropriate.The first paragraph on Page 9: You mentioned that you used a multilevel mixed-effects Poisson regression.  Later on in the paragraph, you mentioned that you applied a mixed-effects logistic regression model. Kindly review and revise the entire paragraph.Results sectionI notice a haphazard presentation of your findings. Your presentation needs to be systematic and must align with the order of arrangement of the factors in the Tables.Page 14: In the first paragraph, you might need to review this statement: "The ICC (Intraclass Correlation Coefficient) value from the null model revealed that only about26.8% of the total variance in stunting among under five children is attributable to differences between clusters, **while the remaining 73.2% was attributed to individual-level factors** ". How was the remaining attributed to individual-level factors alone, and what about the household and contextual factors?Page 15, 1st paragraph: This statement totally negates what is in Table 4: "*The prevalence of stunting was 16% higher (aPR = 1.16, 95% CI: 1.13, 1.19) among children perceived by their mothers as having a small size at birth and 48% higher (aPR = 1.48, 95% CI: 1.43, 1.52) among those perceived as having an average size at birth.* "Page 15, 3rd paragraph: The entire paragraph needs to be rewritten because it is ambiguous, and the findings are lumped together. Kindly use a sentence to present each finding, stating the reference category for each finding.Table 4 has no notation for the asterisk ** (page 18)Discussion section:Page 21, Paragraph 2: Please revise the grammatical structure of this statement: "*Study revealed that, postnatal visit associated with child stunting. Mothers who did not attend postnatal care services were 1.03 more likely to have a stunted child compared to those who did attend postnatal care. This finding consistent with study conducted in Ethiopia(91).* "The discussion lacks coherence. For instance, begin with ANC, then address the Place of delivery, followed by PNC (pages 21-22).Page 22, Paragraph 3, please revise this statements: "*Other factors associated with stunting were single marital status **and household with large family size** . Child stunting higher among mother with single marital status*". **Suggestion** : "*Another factor associated with stunting was marital status. In the study, the prevalence of child stunting was higher among children of single mothers than among children with married mothers* ".Page 22, paragraph 4: "*This study found that, maternal body mass index associated with stunting* ". "**was** " is missing.
plosone@plos.org . A rebuttal letter that responds to each point raised by the academic editor and reviewer(s). You should upload this letter as a separate file labeled 'Response to Reviewers'.A marked-up copy of your manuscript that highlights changes made to the original version. You should upload this as a separate file labeled 'Revised Manuscript with Track Changes'.An unmarked version of your revised paper without tracked changes. You should upload this as a separate file labeled 'Manuscript'.

We look forward to receiving your revised manuscript.

Kind regards,

Ayodeji Babatunde Oginni

Academic Editor

PLOS ONE

Journal Requirements:

https://journals.plos.org/plosone/article?id=10.1371/journal.pone.0248637

https://bmcpregnancychildbirth.biomedcentral.com/articles/10.1186/s12884-024-07105-7

In your revision ensure you cite all your sources (including your own works), and quote or rephrase any duplicated text outside the methods section. Further consideration is dependent on these concerns being addressed.

4. In the online submission form you indicate that your data is not available for proprietary reasons and have provided a contact point for accessing this data. Please note that your current contact point is a co-author on this manuscript. According to our Data Policy, the contact point must not be an author on the manuscript and must be an institutional contact, ideally not an individual. Please revise your data statement to a non-author institutional point of contact, such as a data access or ethics committee, and send this to us via return email. Please also include contact information for the third-party organization, and please include the full citation of where the data can be found.

Reviewers' comments:

Reviewer's Responses to Questions

**Comments to the Author**

1. Is the manuscript technically sound, and do the data support the conclusions?

Reviewer #1: Yes

Reviewer #2: Yes

2. Has the statistical analysis been performed appropriately and rigorously?

Reviewer #1: Yes

Reviewer #2: Yes

3. Have the authors made all data underlying the findings in their manuscript fully available?

Reviewer #1: No

Reviewer #2: Yes

4. Is the manuscript presented in an intelligible fashion and written in standard English?

Reviewer #1: Yes

Reviewer #2: Yes

Reviewer #1: I want to thank the editor and the authors for the chance to review this manuscript. The subject of childhood stunting in sub-Saharan Africa is deeply important, and I appreciate the careful work that has gone into this study. It was a privilege to read and reflect on the findings, and I hope my comments will be useful as you move the paper forward.

General feedback: This paper is very well put together and speaks to a pressing public health concern. The authors draw on DHS data from 35 countries and use appropriate statistical approaches, including robust Poisson regression and mixed-effects modeling, which help account for the survey design and avoid common issues with odds ratios. The sample size is impressive (over 190,000 children), which gives the study strong statistical power and makes the findings more widely applicable. The results are presented clearly, the methods are solid, and the conclusions are well supported by the data.

Introduction

- The introduction does a good job of highlighting the global and regional burden of stunting, but it feels crowded with statistics. The numbers are useful, yet they sometimes overwhelm the main message, which should be that stunting remains a serious child health issue in sub-Saharan Africa despite global progress.

- The authors note that their study fills a gap by using pooled data from 35 countries, but this point could be made more directly. At present, it is not fully clear how this analysis differs from single-country or regional studies and what unique value it adds. Readers should come away with a stronger sense of why this pooled approach is important.

- The introduction leans heavily on prevalence estimates and less on the “why.” It would be stronger if it included more about the pathways linking maternal, household, and structural factors to stunting. This would help readers understand the reasoning behind the choice of variables and provide a clearer foundation for the study. A conceptual framework showing and visualizing pathways would be complementary.

- The section moves quickly between global, regional, and national levels. While this shows the breadth of the issue, the flow could be improved by focusing earlier and more directly on sub-Saharan Africa, since that is the real focus of the paper.

- The writing is generally clear but could benefit from light editing to improve flow and fix awkward wording. For example, phrases like “which previous study not accounted incorporated in to analysis” should be revised for clarity.

Methods

- The paper mentions that 23 variables were included, but it is not entirely clear how these were chosen. While prior studies are referenced, the rationale for why these specific variables were prioritized over others could be explained better. Without that clarity, it is hard to know whether the model might be at risk of being overloaded.

- The manuscript notes that children with missing data were excluded, but it is not clear how the missingness was assessed by the authors or whether its impact on representativeness was considered. Because this is a secondary dataset covering 35 countries, it is possible that patterns in missing data during collection shaped which cases were ultimately included. Adding a short note about this would help bring more clarity and transparency.

- The study draws on surveys from 35 countries collected over more than a decade. Since survey years and tools may vary a bit, it would be helpful to explain how this variation was handled. For example, what steps were taken to make sure that one country or survey year did not carry too much weight in the pooled results?

- The authors mention applying weights, but it is not clear whether these were adjusted for pooling across countries. Since sample sizes vary widely, larger countries may dominate the results if weights are not harmonized. DHS weights make each survey nationally representative, but when pooled they do not automatically align, so rescaling or normalizing is often needed to ensure balanced contributions.

- The authors also report AIC and BIC, but there is no discussion of whether Poisson assumptions were checked. Because stunting data are often overdispersed, ignoring this can bias standard errors and inferences. Robust Poisson regression helps, but a short note on whether overdispersion was considered would add confidence in the analysis.

- Maternal BMI could affect child nutrition, but it could also be shaped by the same household food insecurity or poverty that drives stunting. Similarly, media exposure might influence health behaviors, but access to media is itself determined by socioeconomic status. Without temporal ordering, the relationships in the model can run both ways, which means the results should be interpreted as associations rather than causal effects. Authors should acknowledge the limits of causal interpretation given the cross-sectional design.

Results

- There are places where the confidence intervals seem misordered (for example, lower bounds listed higher than upper bounds). These appear to be typos

- In the country level variation, it is not clear whether differences across countries were statistically tested or whether these patterns are being highlighted simply for descriptive interest. Greater clarity here would help.

- Some of the associations are discussed in a way that suggests causal direction (for example, maternal BMI “reduces risk”), but the results come from a cross-sectional dataset. The narrative would be strengthened by consistently framing these as associations rather than causal effects. For instance, maternal BMI is described as being “associated with stunting,” but the explanation reads as though underweight mothers cause poor nutrient transfer and overweight or obese mothers prevent stunting by having better food access. Media exposure is also presented as an association, yet the discussion suggests media itself changes behaviors and outcomes. In the same way, antenatal and postnatal care are framed as if they directly reduce stunting, with wording that children of mothers who missed visits were “more likely” to be stunted because they missed out on nutrition education.

Discussion

- Although the authors report associations in the results, parts of the discussion read as if the study demonstrates cause and effect. For example, maternal BMI, media exposure, and health service use are described in ways that imply they directly reduce or increase stunting. Because this is a cross-sectional study, the findings should be framed consistently as associations rather than causal effects.

- The discussion does not fully explore the variation across countries. While the pooled analysis is informative, it can mask important differences between contexts. Reflecting on why some countries deviate from the overall pattern, and what that means for interventions, would add more depth.

- Policy implications are mentioned but not clearly prioritized. For instance, maternal education, ANC/PNC attendance, and socioeconomic status are all important, but it would help to distinguish which determinants are more feasible targets for immediate policy action and which require longer-term structural investments.

- The study is cited alongside some prior work, but the discussion could more strongly situate the findings within the broader literature on child stunting in sub-Saharan Africa. How do these results compare with evidence from longitudinal studies or interventions? This would give readers a clearer sense of what is confirmed, what is new, and what remains uncertain.

- The limitations section is rather brief. Issues such as missing data, the cross-sectional design of DHS surveys, and the possibility of residual confounding deserve more explicit acknowledgment. Without this, readers may overestimate the strength of the conclusions.

- Much of the interpretation centers on maternal and household factors. Bringing in structural determinants such as poverty, food systems, governance, or conflict would provide a fuller picture. Without this perspective, the discussion risks placing too much emphasis on individual behavior change as the solution. Think structural and systems-level changes.

Reviewer #2: This is a well-conceived and timely study that addresses a critical public health issue—stunting among under-five children in Sub-Saharan Africa (SSA). The manuscript demonstrates a solid grasp of epidemiological analysis and effectively utilizes secondary data from the Demographic and Health Surveys (DHS). The topic aligns with global priorities such as the Sustainable Development Goals (SDG 2.2), and the methodological rigor is commendable.

**Do you want your identity to be public for this peer review?** For information about this choice, including consent withdrawal, please see our Privacy Policy

Reviewer #1: No

Reviewer #2: No

---

## [Author Response · Author response to Decision Letter 1]

27 Dec 2025

Author response to reviewer

Manuscript ID: PONE-D-25-15249

Manuscript Title: Prevalence of Stunting and its Determinants Among Children Under Five in 35 Sub-Saharan African Countries (2011–2024): Insights from recent Demographic Health Survey Data Using a Generalized Linear Mixed-Effects Model with Robust Poisson Regression

1. Response to Reviewer 1

Comments

Author Response Page number in the manuscript

1. I want to thank the editor and the authors for the chance to review this manuscript. The subject of childhood stunting in sub-Saharan Africa is deeply important, and I appreciate the careful work that has gone into this study. It was a privilege to read and reflect on the findings, and I hope my comments will be useful as you move the paper forward.

2. General feedback: This paper is very well put together and speaks to a pressing public health concern. The authors draw on DHS data from 35 countries and use appropriate statistical approaches, including robust Poisson regression and mixed-effects modeling, which help account for the survey design and avoid common issues with odds ratios. The sample size is impressive (over 190,000 children), which gives the study strong statistical power and makes the findings more widely applicable. The results are presented clearly, the methods are solid, and the conclusions are well supported by the data. Thank you very much for your encouragement and constructive comments.

Introduction

1. The introduction does a good job of highlighting the global and regional burden of stunting, but it feels crowded with statistics. The numbers are useful, yet they sometimes overwhelm the main message, which should be that stunting remains a serious child health issue in sub-Saharan Africa despite global progress. Revised as commented. 3

2. The authors note that their study fills a gap by using pooled data from 35 countries, but this point could be made more directly. At present, it is not fully clear how this analysis differs from single-country or regional studies and what unique value it adds. Readers should come away with a stronger sense of why this pooled approach is important. Revised as commented 3

3. The introduction leans heavily on prevalence estimates and less on the “why.” It would be stronger if it included more about the pathways linking maternal, household, and structural factors to stunting. This would help readers understand the reasoning behind the choice of variables and provide a clearer foundation for the study. A conceptual framework showing and visualizing pathways would be complementary. Revised as commented 5

4. The section moves quickly between global, regional, and national levels. While this shows the breadth of the issue, the flow could be improved by focusing earlier and more directly on sub-Saharan Africa, since that is the real focus of the paper. Revised as commented 3

5. The writing is generally clear but could benefit from light editing to improve flow and fix awkward wording. For example, phrases like “which previous study not accounted incorporated in to analysis” should be revised for clarity. Revised as commented All

Methods

1. The paper mentions that 23 variables were included, but it is not entirely clear how these were chosen. While prior studies are referenced, the rationale for why these specific variables were prioritized over others could be explained better. Without that clarity, it is hard to know whether the model might be at risk of being overloaded. Revised as commented 8-9

2. The manuscript notes that children with missing data were excluded, but it is not clear how the missingness was assessed by the authors or whether its impact on representativeness was considered. Because this is a secondary dataset covering 35 countries, it is possible that patterns in missing data during collection shaped which cases were ultimately included. Adding a short note about this would help bring more clarity and transparency. Revised as commented 8-9

3. The study draws on surveys from 35 countries collected over more than a decade. Since survey years and tools may vary a bit, it would be helpful to explain how this variation was handled. For example, what steps were taken to make sure that one country or survey year did not carry too much weight in the pooled results? Revised as commented 6-7

4. The study draws on surveys from 35 countries collected over more than a decade. Since survey years and tools may vary a bit, it would be helpful to explain how this variation was handled. For example, what steps were taken to make sure that one country or survey year did not carry too much weight in the pooled results? Revised as commented 6

5. The authors mention applying weights, but it is not clear whether these were adjusted for pooling across countries. Since sample sizes vary widely, larger countries may dominate the results if weights are not harmonized. DHS weights make each survey nationally representative, but when pooled they do not automatically align, so rescaling or normalizing is often needed to ensure balanced contributions. Revised as commented 6-7

6. The authors also report AIC and BIC, but there is no discussion of whether Poisson assumptions were checked. Because stunting data are often over dispersed, ignoring this can bias standard errors and inferences. Robust Poisson regression helps, but a short note on whether overdispersion was considered would add confidence in the analysis. Revised as commented 11

7. Maternal BMI could affect child nutrition, but it could also be shaped by the same household food insecurity or poverty that drives stunting. Similarly, media exposure might influence health behaviors, but access to media is itself determined by socioeconomic status. Without temporal ordering, the relationships in the model can run both ways, which means the results should be interpreted as associations rather than causal effects. Authors should acknowledge the limits of causal interpretation given the cross-sectional design. Revised as commented 33

Result

1. There are places where the confidence intervals seem disordered (for example, lower bounds listed higher than upper bounds). These appear to be typos Revised as commented 21-22

2. In the country level variation, it is not clear whether differences across countries were statistically tested or whether these patterns are being highlighted simply for descriptive interest. Greater clarity here would help Revised as commented. We added sensitivity analysis, sub-group and meta-regression for heterogeneity 15-19

3. Some of the associations are discussed in a way that suggests causal direction (for example, maternal BMI “reduces risk”), but the results come from a cross-sectional dataset. The narrative would be strengthened by consistently framing these as associations rather than causal effects. For instance, maternal BMI is described as being “associated with stunting,” but the explanation reads as though underweight mothers cause poor nutrient transfer and overweight or obese mothers prevent stunting by having better food access. Media exposure is also presented as an association, yet the discussion suggests media itself changes behaviors and outcomes. In the same way, antenatal and postnatal care are framed as if they directly reduce stunting, with wording that children of mothers who missed visits were “more likely” to be stunted because they missed out on nutrition education. Revised as commented 21-24

Discussion

1. Although the authors report associations in the results, parts of the discussion read as if the study demonstrates cause and effect. For example, maternal BMI, media exposure, and health service use are described in ways that imply they directly reduce or increase stunting. Because this is a cross-sectional study, the findings should be framed consistently as associations rather than causal effects. Revised as commented 29-30

2. The discussion does not fully explore the variation across countries. While the pooled analysis is informative, it can mask important differences between contexts. Reflecting on why some countries deviate from the overall pattern, and what that means for interventions, would add more depth. Revised as commented 25-26

3. Policy implications are mentioned but not clearly prioritized. For instance, maternal education, ANC/PNC attendance, and socioeconomic status are all important, but it would help to distinguish which determinants are more feasible targets for immediate policy action and which require longer-term structural investments. Revised as commented 33-34

4. The study is cited alongside some prior work, but the discussion could more strongly situate the findings within the broader literature on child stunting in sub-Saharan Africa. How do these results compare with evidence from longitudinal studies or interventions? This would give readers a clearer sense of what is confirmed, what is new, and what remains uncertain. Revised as commented 24-32

5. The limitations section is rather brief. Issues such as missing data, the cross-sectional design of DHS surveys, and the possibility of residual confounding deserve more explicit acknowledgment. Without this, readers may overestimate the strength of the conclusions. Revised as commented 32-33

6. Much of the interpretation centers on maternal and household factors. Bringing in structural determinants such as poverty, food systems, governance, or conflict would provide a fuller picture. Without this perspective, the discussion risks placing too much emphasis on individual behavior change as the solution. Think structural and systems-level changes. Revised as commented 24-32

2. Response to Reviewer 2

Comments

Author Response Page number in the manuscript

1. Thank you for submitting your manuscript to PLOS ONE. After careful consideration, we feel that it has merit but does not fully meet PLOS ONE’s publication criteria as it currently stands. Therefore, we invite you to submit a revised version of the manuscript that addresses the points raised during the review process.

2. This is a well-conceived and timely study that addresses a critical public health issue—stunting among under-five children in Sub-Saharan Africa (SSA). The manuscript demonstrates a solid grasp of epidemiological analysis and effectively utilizes secondary data from the Demographic and Health Surveys (DHS). The topic aligns with global priorities such as the Sustainable Development Goals (SDG 2.2), and the methodological rigor is commendable. Thank you very much for your encouragement and constructive comments.

General comments

1. You did not line-number your manuscript to help reviewers pinpoint the exact areas requiring revision in the manuscript. Revised as commented All

2. There is inconsistent use of terms for your outcome variable: "stunting among U-5 children", "stunting", "childhood stunting", "stunting among under five children", Revised as commented All

Abstract

1. Your abstract exceeds 300 words, so you need to be parsimonious in the reporting of your findings. You can report the most salient findings: the child's sex, age & birth size; the mother's education; and household wealth, i.e, those with aPR ≥±20% Revised as commented 2-3

2. Please revise this statement "Key variables such as antenatal care visits, postnatal care, and maternal nutritional indicators which previous study not accounted incorporated in to analysis". Suggestion: Key variables such as antenatal care visits, postnatal care, and maternal nutritional indicators, which previous studies did not account for, are incorporated into the analysis. Revised as commented 2

3. Please revise this statement: "The adjusted prevalence ratio and 95% confidence intervals were used to indicate statistical significance and the strength of associations". Suggestion: "The adjusted prevalence ratios and their 95% confidence intervals were used to assess the strength and statistical significance of associations". Revised as commented 2

4. The abstract results are overloaded with too much information. Kindly reduce the results to the salient ones as suggested above. In fact, the entire multivariate analysis results in the result section need to be rewritten. Again, the term "children older age" is ambiguous, and the aPR value of 2.32, which you stated, is misleading. It is better to say: "being aged 12 months or older" (aPR: ≥1.81, p<0.01). Also, "mother without a higher education (aPR: ≥1.68, p<0.01), and living in a poor or average household (aPR: ≥1.23, p<0.01). Revised as per the comments. We tried as much as possible. 2

Methods

1. I suggest you merge the poorest and the poorer categories into one category, "poor", and merge the richer and the richest into one category as "rich", thus you will have three categories for the wealth index. This will provide a much better explanation of your findings. Revised as commented 10

2. The 2nd paragraph on Page 8: I see that you included "source of drinking water", "type of toilet facility", and "type of cooking" in your independent variable list. I suggest you exclude these variables because they have already been used to estimate the household wealth index (composite index). Using them again with the household wealth index in the model may not be statistically appropriate. Revised as commented 10

3. The first paragraph on Page 9: You mentioned that you used a multilevel mixed-effects Poisson regression. Later on in the paragraph, you mentioned that you applied a mixed-effects logistic regression model. Kindly review and revise the entire paragraph. Revised as commented 9-10

Result

1. I notice a haphazard presentation of your findings. Your presentation needs to be systematic and must align with the order of arrangement of the factors in the Tables. Revised as commented 21-24

2. Page 14: In the first paragraph, you might need to review this statement: "The ICC (Intraclass Correlation Coefficient) value from the null model revealed that only about 26.8% of the total variance in stunting among under five children is attributable to differences between clusters, while the remaining 73.2% was attributed to individual-level factors". How was the remaining attributed to individual-level factors alone, and what about the household and contextual factors? Revised as commented 20-21

3. Page 15, 1st paragraph: This statement totally negates what is in Table 4: "The prevalence of stunting was 16% higher (aPR = 1.16, 95% CI: 1.13, 1.19) among children perceived by their mothers as having a small size at birth and 48% higher (aPR = 1.48, 95% CI: 1.43, 1.52) among those perceived as having an average size at birth." Revised as commented All pages

4. Page 15, 3rd paragraph: The entire paragraph needs to be rewritten because it is ambiguous, and the findings are lumped together. Kindly use a sentence to present each finding, stating the reference category for each finding. Revised as commented 21-24

5. Table 4 has no notation for the asterisk ** (page 18) Revised as commented 24

Discussion section

1. Page 21, Paragraph 2: Please revise the grammatical structure of this statement: "Study revealed that, postnatal visit associated with child stunting. Mothers who did not attend postnatal care services were 1.03 more likely to have a stunted child compared to those who did attend postnatal care. This finding consistent with study conducted in Ethiopia (91)." Revised as commented 30

2. The discussion lacks coherence. For instance, begin with ANC, then address the Place of delivery, followed by PNC (pages 21-22). Revised as commented 21-24

3. Page 22, paragraph 4: "This study found that, maternal body mass index associated with stunting". "was" is missing. Revised as commented

3. Response to Journal Requirements:

Comments

Author Response Page number in the manuscript

1. Please ensure that your manuscript meets PLOS ONE's style requirements, including those for file naming. The PLOS ONE style templates can be found at: https://journals.plos.org/plosone/s/file?id=wjVg/PLOSOne_formatting_sample_main_body.pdf and https://journals.plos.org/plosone/s/file?id=ba62/PLOSOne_formatting_sample_title_authors_affilia

---

## [Editor Report · Decision Letter 1]

1 Jan 2026

Dear Dr. Moloro,

We look forward to receiving your revised manuscript.

Kind regards,

Ayodeji Babatunde Oginni

Academic Editor

PLOS One
---

## [Author Response · Author response to Decision Letter 2]

15 Feb 2026

Author response to reviewer

Manuscript ID: PONE-D-25-15249R1

Manuscript Title: Prevalence of Stunting and its Determinants Among Children Under Five in 35 Sub-Saharan African Countries (2011–2024): Insights from recent Demographic Health Survey Data Using a Generalized Linear Mixed-Effects Model with Robust Poisson Regression

1. Response to Reviewer 1

Comments

Author Response Page number in the manuscript

1. Thank you for submitting your manuscript to PLOS ONE. After careful consideration, we feel that it has merit but does not fully meet PLOS ONE’s publication criteria as it currently stands. Therefore, we invite you to submit a revised version of the manuscript that addresses the points raised during the review process. Thank you very much for your encouragement and constructive comments.

Typographic and grammatical issues

1. After a careful review of the revised manuscript, a few typographic and grammatical issues were identified and need to be addressed. They are highlighted with comments or suggestions in the attached document for your action. They are located in the following lines: 49-51; 61-64; 126; 306-307 (Table 2); 401-403; 431-432; 470; 495; 532-533; 551-552; 554-556; 560-561; 566-567; 631-632; 643; 707; 708; 709; 711; 719; 726; & 729. All are revised as commented.

2. Response to Journal Requirements:

Comments

Author Response: Page number in the manuscript

1. If the reviewer comments include a recommendation to cite specific previously published works, please review and evaluate these publications to determine whether they are relevant and should be cited. There is no requirement to cite these works unless the editor has indicated otherwise. Evaluated as commented.

2. Please review your reference list to ensure that it is complete and correct. If you have cited papers that have been retracted, please include the rationale for doing so in the manuscript text, or remove these references and replace them with relevant current references. Any changes to the reference list should be mentioned in the rebuttal letter that accompanies your revised manuscript. If you need to cite a retracted article, indicate the article’s retracted status in the References list and also include a citation and full reference for the retraction notice. No retracted paper has been cited as much as we evaluated.

2. You may also use PLOS’s free figure tool, NAAS, to help you prepare publication-quality figures: https://journals.plos.org/plosone/s/figures#loc-tools-for-figure-preparation.

Revised as commented. All

---

## [Editor Report · Decision Letter 2]

20 Feb 2026

Prevalence of Stunting and its Determinants Among Children Under Five in 35 Sub-Saharan African Countries (2011–2024): Insights from recent Demographic Health Survey Data Using a Generalized Linear Mixed-Effects Model with Robust Poisson Regression

PONE-D-25-15249R2

Dear Dr. Moloro,

We’re pleased to inform you that your manuscript has been judged scientifically suitable for publication and will be formally accepted for publication once it meets all outstanding technical requirements.

Kind regards,

Ayodeji Babatunde Oginni

Academic Editor

PLOS One
---

## [Editor Report · Acceptance letter]

PONE-D-25-15249R2

PLOS One

Dear Dr. Moloro,

I'm pleased to inform you that your manuscript has been deemed suitable for publication in PLOS One. Congratulations! Your manuscript is now being handed over to our production team.

Kind regards,

on behalf of

Ayodeji Babatunde Oginni

Academic Editor

PLOS One